# Causal Discovery in Linear Structural Causal Models with Deterministic Relations

**Yuqin Yang**                                         YUQINYANG@GATECH.EDU
*School of Mathematics, Georgia Institute of Technology,*
*Atlanta, GA 30332, USA*

**Mohamed Nafea**                                      NAFEAMO@UDMERCY.EDU
*Electrical & Computer Engineering Department, University of Detroit Mercy,*
*Detroit, MI 48221, USA*

**AmirEmad Ghassami**                                  AGHASSA1@JHU.EDU
*Department of Computer Science, Johns Hopkins University,*
*Baltimore, MD 21218, USA*

**Negar Kiyavash**                                     NEGAR.KIYAVASH@EPFL.CH
*College of Management of Technology, École Polytechnique Fédérale de Lausanne (EPFL),*
*Lausanne, Switzerland*

**Editors:** Bernhard Schölkopf, Caroline Uhler and Kun Zhang

## Abstract

Linear structural causal models (SCMs)– in which each observed variable is generated by a subset of the other observed variables as well as a subset of the exogenous sources– are pervasive in causal inference and casual discovery. However, for the task of causal discovery, existing work almost exclusively focus on the submodel where each observed variable is associated with a distinct source with non-zero variance. This results in the restriction that no observed variable can deterministically depend on other observed variables or latent confounders. In this paper, we extend the results on structure learning by focusing on a subclass of linear SCMs which do not have this property, i.e., models in which observed variables can be causally affected by any subset of the sources, and are allowed to be a deterministic function of other observed variables or latent confounders. This allows for a more realistic modeling of influence or information propagation in systems. We focus on the task of causal discovery form observational data generated from a member of this subclass. We derive a set of necessary and sufficient conditions for unique identifiability of the causal structure. To the best of our knowledge, this is the first work that gives identifiability results for causal discovery under both latent confounding and deterministic relationships. Further, we propose an algorithm for recovering the underlying causal structure when the aforementioned conditions are satisfied. We validate our theoretical results both on synthetic and real datasets.

**Keywords:** Causal Discovery, Structural Causal Models, Deterministic Relations, Blind Source Separation

## 1. Introduction

Causal discovery, which refers to the problem of learning causal relationships among the variables of a system, has received extensive attention with applications ranging from social sciences, economics, all the way to biology. The gold standard of causal discovery is performing interventions (i.e., controlled experiments). But such experiments could be costly, infeasible, or even at times unethical. This necessitates developing statistical methods based on purely observational data, where further assumptions on the data generating process are needed.

Linear structural causal models (SCMs) have been extensively considered in the literature perhaps as the most pervasive causal data generating model (Pearl, 2009; Spirtes et al., 2000; Peters et al., 2017). In this model, the system is comprised of a set of observed (endogenous) variables and a set of source (exogenous) variables. Each observed variable $x$ is generated as a linear combination of a subset of the other observed variables $Pa(x)$ (called direct causes of $x$) plus a function of a subset of the source variables $S(x)$. We refer to this model as the linear general SCM (G-SCM).

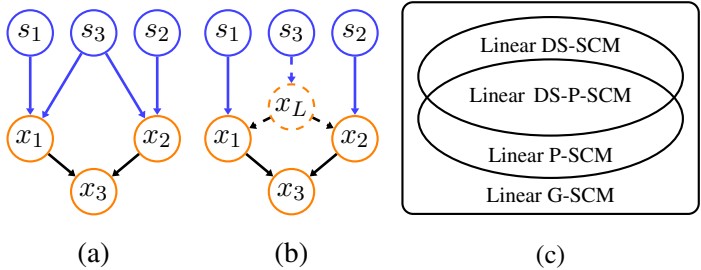

(a)      (b)      (c)

Figure 1: (a) A linear P-SCM with observed variables $x_1, x_2, x_3$ and independent sources $s_1, s_2, s_3$. This model cannot be represented as a linear DS-SCM, since $x_3$ is fully determined by $(x_1, x_2)$. (b) An interpretation of the P-SCM in (a), where $x_1$, $x_2$ are both influenced by a latent confounder $x_L$. (c) Relations among linear causal models. The intersection DS-P-SCM represents the linear causal models under distinct source assumption, jointly independent sources and linear latent confounding.

Literature on linear G-SCM for causal discovery almost exclusively focuses on a subset of G-SCM, in which for each observed variable $x$, at least one source in $S(x)$ is distinct and has non-zero variance (Peters et al., 2017). Here, a distinct source of observed variable $x$ means a source $s \in S(x)$ that does not belong to any other subsets $S(x')$, $x' \neq x$. Then the non-distinct (i.e., shared) sources can be explained as *latent confounders* in the system. We refer to this submodel of G-SCM as *distinct source SCM* (DS-SCM). This restriction implies that no observed variable can be a deterministic function of any other observed variables and/or any latent confounders in the system. However, in a G-SCM, an observed variable is allowed to have no distinct sources (or equivalently can have a distinct source with zero-variance). Hence, deterministic relations are allowed. Deterministic relations are problematic in causal discovery as they introduce additional dependencies among variables (Peters et al., 2017). Specifically, when deterministic relations exist, d-separation is not complete anymore (although it remains sound). Such complications are perhaps the reason that deterministic relations are avoided for the most part in the literature.

Furthermore, G-SCMs without distinct sources can have practical significance in problems beyond causal modelling. This subset of G-SCMs can be used to model influence propagation through networks, where the observed nodes are influenced directly or indirectly by a set of external factors (sources). Examples of such factors include sources of news or rumors, or sources of diseases. Specifically, G-SCMs without distinct sources can be used to model scenarios with less factors than observed nodes; this situation arises when a few sources influence a group of observed variables in the system. For instance, in a setting of spread of news, a large group of individuals may have their news exclusively from a limited set of news sources (Guille et al., 2013; Quinn et al., 2015). Other examples include spread of epidemics (Nowzari et al., 2016), political campaigns (González-Bailón and Wang, 2016), message passing in human brain networks (Karwowski et al., 2019).

In this paper, we focus on another subclass of linear G-SCMs which relaxes the distinct source assumption of DS-SCM and hence, allows for deterministic relations among the observed variables. In our proposed model, we consider a set of jointly independent source variables. Each observed

variable is generated as a linear combination of the other observed variables plus a linear combination of the sources. We relax the distinct source assumption of DS-SCM by assuming instead that each observed variable contains strictly more sources than any of its direct causes. Figure 1(a) depicts an example which satisfies our assumption but violates the assumption of DS-SCM. As previously mentioned, sharing sources introduces latent confounding among the variables. Hence our introduced model allows for both latent confounders as well as deterministic relations. We refer to this model as the linear *propagation SCM* (P-SCM). Majority of works in the literature on linear causal models in fact consider an intersection of P-SCM and DS-SCM, which we refer to as DS-P-SCM. Therefore, our work strictly expands the considered model space compared to those works. See Figure 1(c) for a schematic representation of the relations among the linear models.

We study the problem of causal discovery when the data generating model is a linear P-SCM, that is, to learn the directions and strengths of the causal influences among the observed variables, as well as the direct connections from the sources to the observed variables. We derive a set of necessary and sufficient conditions for unique identifiability of the causal structure from observational data under the assumptions of faithfulness and separability of the sources. To the best of our knowledge, this is the first work that gives identifiability results for causal discovery under both latent confounding and deterministic relationships. We further propose an algorithm which uniquely identifies the causal structure when the conditions are satisfied. Also, we derive an equivalent condition when the generating model is a linear DS-P-SCM. We examine our algorithm on synthetic and real datasets, and compare with existing causal discovery methods developed for linear DS-SCMs with or without latent confounders. The results show that our proposed algorithm outperforms those methods in recovering the underlying structure.

**Related work.** Under causal sufficiency, Shimizu et al. (2006) showed that a linear non-Gaussian additive model (LiNGAM), in which source variables have non-Gaussian distributions, allows for unique identifiability of the causal structure from observations. Causal sufficiency is a common assumption in causal discovery which means there exist no latent confounders for any pair of observed variables. Yet, without considering the effect of latent confounders, one may infer wrong causal relations among observed variables. Hoyer et al. (2008) and Salehkaleybar et al. (2020) considered an extension of LiNGAM in the presence of latent variables (lvLiNGAM). Our model generalizes lvLiNGAM to the case when observed variables are allowed to deterministically depend on other observed variables or latent confounders. We further comment on the connection of our model to lvLiNGAM in Appendix C.4.

SCMs with deterministic relations have been considered in a few work. In (Geiger et al., 1990; Spirtes et al., 2000), D-separation condition (with capital letter D) is proposed for graphically determining conditional independencies when deterministic relations are allowed. Yet, it remained unclear whether D-separation condition can capture *all* conditional independencies induced from the distribution. Daniušis et al. (2010) and Janzing et al. (2012) considered a system with only two variables, one variable deterministically causing the other, and showed that the correct causal direction can be learnt if the deterministic function contains no information about the cause variable and vice versa. Their analysis does not hold for linear relations though. Scheines et al. (1996) considered recovering a reduced model in which all deterministic variables are removed. Luo (2006); Mabrouk et al. (2014); Lemeire et al. (2012) adapted conventional causal discovery methods such as PC and greedy-search algorithms (Pearl, 2009; Tsamardinos et al., 2006; Chickering, 2002), where deterministic relations are detected either by additional independence tests, or by calculating conditional entropy among variables. The aforementioned methods suffer from identifiability problem.

In particular, majority of these works do not discuss the capability of identifying the underlying structure by their proposed algorithms (e.g., the equivalence classes of the recovery result). Besides, these methods do not consider the presence of latent confounders in the system. Please refer to Appendix A for a detailed background on causal discovery methods.

## 2. System model

*Notation.* We use upper-case letters for vectors and bold upper-case letters for matrices. We use $[n] \triangleq \{1, 2, \cdots, n\}$. If $X = [x_1, \cdots, x_p]$, $X_{[i:j]}$ is the sub-vector $[x_i, x_{i+1}, \cdots, x_j]$; $1 \leq i < j \leq p$. We use $x_i \rightsquigarrow x_j$ to refer to a directed path from node $x_i$ to node $x_j$ (in the corresponding graph).

### 2.1. Generating model

As we mentioned in the introduction, linear G-SCM is one of the main data generating models assumed in the literature. A G-SCM (Pearl, 2009, Definition 7.1.1) is a 4-tuple $\langle \mathcal{S}, \mathcal{X}, \mathcal{F}, P(\mathcal{S}) \rangle$. $\mathcal{S}$ is a set of source variables, $\mathcal{X}$ is a set of observed variables, $\mathcal{F} = \{f_x\}_{x \in \mathcal{X}}$ is a set of functions such that for each $x \in \mathcal{X}$, $x = f_x(Pa(x), S(x))$, $Pa(x) \subseteq \mathcal{X} \setminus \{x\}$ and $S(x) \subseteq \mathcal{S}$, and $P(\mathcal{S})$ is a joint probability distribution over $\mathcal{S}$. In the case of linear functions, the model is restricted as follows:

**Definition 1 (Linear G-SCM)** *A linear G-SCM is the subclass of G-SCM in which each observed variable $x$ is generated as $x = f_x(Pa(x)) + g_x(S(x))$, where $f_x$ is a linear function and $g_x$ is a function indicating the effects from the sources. We refer to $g_x(S(x))$ as the source mixture corresponding to observed variable $x$.*

Previous works on causal discovery mostly focus on a subclass of linear G-SCM which satisfies the distinct source assumption defined below.

**Assumption 1 (Distinct source assumption)** *A linear G-SCM satisfies the distinct source assumption if each $S(x)$ contains at least one distinct source, that is, a source with non-zero variance that does not appear in any other $S(x')$ for $x' \neq x$.*

We refer to the subclass of linear G-SCMs satisfying Assumption 1 as linear DS-SCM. The non-distinct (i.e., shared) sources represent latent confounders in the system. Assumption 1 can be interpreted as: The source mixture of one variable cannot be a deterministic function of the components in the source mixtures of the rest of the variables.[1] Equivalently, no observed variable can be a deterministic function of any other observed variables and/or any latent confounders in the system.

In this paper, we introduce another subclass of linear G-SCMs, which we refer to as the linear propagation SCM (P-SCM). Consider a set of observed variables $\mathcal{X} = \{x_1, \cdots, x_p\}$ arranged in a causal order (no latter variable causes any earlier variable), and another set of jointly independent source variables $\mathcal{S} = \{s_1, \cdots, s_m\}$. In the generating model of a linear P-SCM, each observed variable consists of a linear combination of earlier observed variables plus another linear combination of the sources. Specifically,

$$x_i = \sum_{j=1}^{i-1} a_{ij} x_j + \tilde{s}_i, \qquad \tilde{s}_i = g_{x_i}(S(x_i)) = \sum_{j=1}^{m} b_{ij} s_j. \tag{1}$$

$a_{ij}$ is the strength of the direct causal effect from $x_j$ to $x_i$. $\tilde{s}_i$, $i \in [p]$ represents the mixture of sources exogenously causing $x_i$. $b_{ij}$ is the strength of the direct exogenous effect from $s_j$ to $x_i$. We refer to $\{\tilde{s}_1, \cdots, \tilde{s}_p\}$ as *source mixtures* or simply *mixtures*. Note that since (1) allows observed variables to share sources, our model allows for latent confounding.[2] However, the latent confounding can

---

1. In fact, Assumption 1 is stronger than what is needed for structure learning methods developed for linear DS-SCMs. We provide a weaker version of Assumption 1 in Appendix D.1, under which these methods still work.

2. Note that prohibiting latent confounders, i.e., causal sufficiency, requires independent source mixtures.

only be linear, i.e., it is due to the source mixtures being linear combinations of elements in $\mathcal{S}$. In our model we require the following assumption on the generating model which is weaker than Assumption 1 of DS-SCM.

**Assumption 2**   *If $x_i$ is a direct cause of $x_j$, then there is at least one source in $x_j$ that is not in $x_i$.*

Unlike the distinct source assumption in DS-SCMs, Assumption 2 does not necessarily require each observed variable to be associated with a distinct source (see Figure 1(a) for an example). Therefore, our model allows for deterministic relations.[3] We provide a detailed comparison of our model with DS-SCM in Section 4.

We define the causal diagram of the model as a directed graph in which the vertex set consists of the observed variables $\mathcal{X}$ and the source variables $\mathcal{S}$. There is a directed edge from observed variable $x_i$ to $x_j$ if and only if $a_{ji} \neq 0$ (we refer to such an edge as a causal connection), and a directed edge from source $s_i$ to observed variable $x_j$ if and only if $b_{ji} \neq 0$ (we refer to such an edge as an exogenous connection). Due to the assumed causal order, the causal diagram is acyclic, and hence the model is a directed acyclic graph (DAG). The model in (1) can be written in the matrix form as

$$X = \mathbf{A}X + \tilde{S}; \qquad \tilde{S} = \mathbf{B}S, \tag{2}$$

where $X \triangleq [x_1, \cdots, x_p]^\top$, $\tilde{S} \triangleq [\tilde{s}_1, \cdots, \tilde{s}_p]^\top$, $S \triangleq [s_1, \cdots, s_m]^\top$. $\mathbf{A}$ is a $p \times p$ strictly lower triangular matrix with the coefficient $a_{ij}$ on the $(i, j)$-th entry. $\mathbf{B}$ is a $p \times m$ matrix with the coefficient $b_{ij}$ on the $(i, j)$-th entry. The strictly lower triangular matrix $\mathbf{A}$ attests the existence of a causal order among the observed variables (the generating process is recursive (Bollen, 1989)). Mixtures $\tilde{s}_i$, $i \in [p]$ represent *additive exogenous noises*.

## 2.2. Identifiability assumptions

In (2), each $x_i$ can be written as a linear combination of the sources. Let $\mathbf{W}$ be a mixing matrix s.t.

$$X = \mathbf{W}S, \qquad \text{where} \quad \mathbf{W} = (\mathbf{I} - \mathbf{A})^{-1}\mathbf{B}, \tag{3}$$

Define the component set of a variable $x_i = \sum_j w_{ij}s_j$ (or a mixture $\tilde{s}_i = \sum_j b_{ij}s_j$) as $\mathrm{Comp}(x_i) = \{s_j \in \mathcal{S} \,|\, w_{ij} \neq 0 \text{ (or } b_{ij} \neq 0)\}$. We require the following assumptions for identifiability.

**Assumption 3 (P-SCM faithfulness)**
*(a) If $x_i$ is an ancestor of $x_j$, then all source components in $x_i$ must also appear in $x_j$.*
*(b) Let $\mathbf{B}_p$ be an arbitrarily permuted version of matrix $\mathbf{B}$ (either row or column permutations). Any submatrix of $\mathbf{B}_p$ (with non-zero rows or columns) is of full rank.*

**Assumption 4 (Separability)**   *We assume the linear P-SCM is separable– its corresponding mixing matrix $\mathbf{W}$, cf. (3), can be correctly recovered using observations $X$, up to scaling and permutation of its columns.*

P-SCM faithfulness assumption is extended from the common faithfulness assumption in studying causal models (Spirtes et al., 2000; Pearl, 2009). This assumption is satisfied *almost surely* if all model coefficients $\{a_{ij}\}$ and $\{b_{ij}\}$ are drawn randomly and independently from continuous distributions (Meek, 1995). Separability, on the other hand, can be achieved using blind source separation (BSS) methods (Comon and Jutten, 2010), given certain assumption on the sources. For example, when the

---

3. If the linear P-SCM only consists of two observed variables $x_i, x_j$, then Assumption 2 implies that $x_j$ cannot be a deterministic function of $x_i$ and shared sources. This example is however a degenerate case. In particular, given a deterministic relation between $x_i$ and $x_j$, it is not possible to recover their causal direction based on observations.

sources are non-Gaussian random variables, the mixing matrix $\mathbf{W}$ can be recovered using Independent Component Analysis (ICA) or overcomplete ICA methods (Comon, 1994; Hyvarinen et al., 2002; Lewicki and Sejnowski, 2000). Other BSS methods include Statistical Blind Source Separation Regression (SBSSR) model and Non-negative Matrix Factorization (NMF). See Appendices B and C for detailed explanation about both assumptions, and how BSS methods are applied to our model. Note that the result of BSS may suffer from permutation and scalability indeterminacies, due to lack of prior information about the sources. In Appendix C.4 we show that these indeterminacies do not affect the recovery performance of our algorithm. Specifically, we show that, even in the presence of these indeterminacies, our recovered model is identical to the true model.

## 3. Conditions for unique identifiability

In this section, we present the necessary and sufficient conditions for unique identifiability of a linear P-SCM, which consists of two parts, the unique components condition and marriage condition. Both conditions are imposed on the "possible parent set" of each observed variable which is a superset of its ancestors that can be recovered from the mixing matrix (from observational data). We first define the possible parent set, and then present the combinatorial version of the conditions. Equivalent algebraic (matrix) representations of the conditions are provided in Appendix E.1.

The necessary and sufficient condition we derive for unique identifiability of a linear P-SCM, as well as our proposed algorithm, require a search over the *possible parent set* of an observed variable $x_k$ which is defined as follows.

**Definition 2** *The* possible parent set *of an observed variable $x_k$ is defined as the set of observed variables in $\mathcal{X}$ whose component set is a strict subset of the component set of $x_k$. That is, $\mathcal{P}_k = \{x_i \in \mathcal{X} | \mathrm{Comp}(x_i) \subsetneq \mathrm{Comp}(x_k)\}$.*

We refer to the elements of this set as the "possible parents" of $x_k$ since each element is a candidate to be a parent of $x_k$, according to P-SCM faithfulness assumption. The possible parent set of $x_k$ (i) can be deduced from the mixing matrix and (ii) can be strictly smaller than the set of observed variables preceding $x_k$ in the causal order; see Appendix D.3 for an example.

In linear DS-SCMs, since each observed variable has a distinct source, $\mathcal{P}_k$ only includes the ancestors of $x_k$ in the generating model (see Section 4.1 for more explanation). This property allows to identify all the observable ancestors of each variable $x_k$ from the mixing matrix. Subsequently, an algorithm, such as lvLiNGAM in (Salehkaleybar et al., 2020), can recover the directed causal paths among variables. By contrast, in our linear P-SCM, the possible parent set $\mathcal{P}_k$ can include observed variables which are not ancestors of $x_k$. The reason behind this is that there may exist an observed variable which does not have a distinct source, hence this variable can mimic an ancestor of $x_k$ by its exogenous connections to the sources.

From (3), the observed variables are given by $X = (\mathbf{I} - \mathbf{A})^{-1}\mathbf{B}S = (\mathbf{I} - \mathbf{A})^{-1}\tilde{\mathbf{B}}S'$, where $\tilde{\mathbf{B}}$ is a version of $\mathbf{B}$ with the columns arbitrarily permuted and rescaled such that $\tilde{\mathbf{B}}S' = \mathbf{B}S$. Note that the $(i, j)$-th entry of $(\mathbf{I} - \mathbf{A})^{-1}$ represents the *total causal effect* from observed variable $x_j$ to observed variable $x_i$ (Spirtes et al., 2000). Our task is to uniquely recover $(\mathbf{I} - \mathbf{A})^{-1}$, i.e., the total causal effects among observed variables, and to recover $\mathbf{B}$ up to permutation and scaling of its columns, i.e., the exogenous connections from the sources to the observed variables. Note that the permutation indeterminacy of $\mathbf{B}$ is immaterial since there exists no ordering between the exogenous connections, and we are not concerned about the scalability indeterminacy in recovering their effect. Next, we introduce the necessary and sufficient conditions for recovering $(\mathbf{I} - \mathbf{A})^{-1}$ and $\tilde{\mathbf{B}}$.

### 3.1. Unique components condition

When recovering the total causal effect from possible parents of $x_k$ (i.e., variables in $\mathcal{P}_k$) to $x_k$, the question is whether each existing component in $x_k$ (i.e., $s \in \bigcup_{x_i \in \mathcal{P}_k} \mathrm{Comp}(x_i)$) comes from a possible parent ($x_i \in \mathcal{P}_k$), or comes from the exogenous connections to $x_k$ (i.e., $\tilde{s}_k = \sum_{l=1}^{m} b_{kl} s_l$), or maybe both. To address this question, we propose the concept of *unique components*. For a possible parent $x_i \in \mathcal{P}_k$, if $\tilde{s}_i$ has a component that is not shared by the source mixture of any other possible parent in $\mathcal{P}_k$ (i.e., any $\tilde{s}_j$; $j \neq i$, $x_j \in \mathcal{P}_k$), then we can use this *unique component* to compute the total causal effect from $x_i$ to $x_k$ (by dividing the coefficient of this unique component in $x_k$ by its coefficient in $x_i$, see Remark 9 in the sufficiency proof for more details).

We can recursively use the unique components idea to find the total causal effects from all possible parents in $\mathcal{P}_k$ to $x_k$: After using the unique components to recover total causal effects from the corresponding possible parents to $x_k$, we remove those variables and search for unique components with respect to the remainder subset of $\mathcal{P}_k$. Thus, we extend the concept of unique components to subsets of $\mathcal{P}_k$ by considering only the possible parents in these subsets. We define the unique components over the mixtures $\tilde{\mathcal{S}}_k = \{\tilde{s}_i : x_i \in \mathcal{P}_k\}$ in an iterative manner as follows.

**Definition 3** *Fix an observed variable $x_k$. For each possible parent of $x_k$, i.e., $x_i \in \mathcal{P}_k$, we define the* unique component set *of $x_i$, denoted by $U_k(i)$, as follows.*
*(i) Let $J^{(0)} = \{i : x_i \in \mathcal{P}_k\}$, i.e., the index set of the possible parent set of $x_k$.*
*(ii) For $n = 1, 2, \cdots$ ($n$ is the iteration index):*
  *- For $i \in J^{(n-1)}$, define $\tilde{U}_k^{(n)}(i) = \mathrm{Comp}\left(\tilde{s}_i\right) \setminus \bigcup_{j \in J^{(n-1)}, j \neq i} \mathrm{Comp}\left(\tilde{s}_j\right).$*
  *- Let $J^{(n)} = \{i \in J^{(n-1)} | \tilde{U}_k^{(n)}(i) = \emptyset\}$.*
  *- Repeat until $|J^{(N+1)}| = |J^{(N)}|$ for some $N \in \mathbb{N}$. Define $\mathcal{I}_k = \{x_l : l \in J^{(N)}\}$.*
*For each $x_i \in \mathcal{P}_k$, $U_k(i) = \tilde{U}_k^{(n)}(i)$, $i \in J^{(n-1)} \setminus J^{(n)}$, $n = 1, \cdots, N$; $U_k(i) = \emptyset$, $i \in J^{(N)}$.*

In Definition 3, when $n = 1$, $\tilde{U}_k^{(1)}(i)$ represents the components in $\tilde{s}_i$ but not in any other $\tilde{s}_j$, where $x_j \in \mathcal{P}_k$ (or equivalently $j \in J^{(0)}$). $J^{(1)}$ represents the index set of $\mathcal{P}_k^{(1)}$ which is the subset of $\mathcal{P}_k$ composed of observed variables with no unique components in the first iteration. In the next iteration, $\tilde{U}_k^{(2)}(i)$ is defined similarly but only for $i \in J^{(1)}$. $J^{(2)}$ is the index set of the subset of $\mathcal{P}_k^{(1)}$ with no unique components. This procedure is repeated until no more unique components are found; $\mathcal{I}_k \subseteq \mathcal{P}_k$ is the remaining subset of possible parents with no unique components after the procedure terminates. Appendix D.4 provides an example to demonstrate this iterative definition.

We use Definition 3 as follows. We (i) identify the possible parents in $\mathcal{P}_k$ with unique components, (ii) compute their total causal effects to $x_k$ using these unique components, and (iii) subtract the learnt causal effects from $x_k$. Next, we consider only the possible parents with no unique components in the first iteration ($x_i \in \mathcal{P}_k^{(1)}$), and find which of these have unique components among variables in $\mathcal{P}_k^{(1)}$. We then use these unique components to compute total causal effects from the corresponding variables in $\mathcal{P}_k^{(1)}$ to $x_k$. Based on Definition 3, we state the unique components condition as follows.

**Condition 1 (Unique components condition)** *(i) For each possible parent $x_i$ of $x_k$ (i.e., $x_i \in \mathcal{P}_k$) with a non-empty unique component set (i.e., $|U_k(i)| \neq 0$), the unique components of $x_i$ are not exogenously connected to $x_k$. That is, $\mathrm{Comp}(\tilde{s}_k) \cap U_k(i) = \emptyset$, $\forall x_i \in \mathcal{P}_k$ s.t. $U_k(i) \neq \emptyset$.*
*(ii) Further, for any $x_i \in \mathcal{P}_k$ with an empty unique component set ($|U_k(i)| = 0$), the exogenous connections to $x_i$ and $x_k$ are disjoint. That is, $\mathrm{Comp}(\tilde{s}_k) \cap \mathrm{Comp}(\tilde{s}_i) = \emptyset$, $\forall x_i \in \mathcal{P}_k$ s.t. $U_k(i) = \emptyset$.*

---

**Algorithm 1:** P-SCM Recovery

---

**Input:** Recovered mixing matrix $\tilde{\mathbf{W}}$ from BSS.   **Initialize:** $\tilde{\mathbf{A}} = \mathbf{I}$, $\mathbf{B} = \mathbf{0}$.

1  Repermute $\tilde{\mathbf{W}}$ such that the number of non-zero entries in each row is in an increasing order
  (rows with equal number of non-zero entries are permuted at random) ;

2  **for** $k = 1 : p$ **do**

3      Find possible parent set $\mathcal{P}_k$ using $\tilde{\mathbf{W}}$; Select the row $W_k = \tilde{\mathbf{W}}[k,:]$; Initialize $\bar{\mathcal{P}}_k = \mathcal{P}_k$ ;

4      Find the set $\mathcal{U}$ of possible parents in $\bar{\mathcal{P}}_k$ that have unique components ;

5      **if** $|\mathcal{U}| \neq 0$ **then**

6          Compute the total causal effect $\tilde{a}_{ki}$ for each $x_i \in \mathcal{U}$ using its unique components ;

7          $W_k \leftarrow W_k - \sum_{x_i \in \mathcal{U}} \tilde{a}_{ki}\mathbf{B}[i,:]$; $\bar{\mathcal{P}}_k \leftarrow \bar{\mathcal{P}}_k \setminus \mathcal{U}$; Go back to Step 4 using updated $\bar{\mathcal{P}}_k$ ;

8      **else**

9          Select the rows of $\mathbf{B}$ corresponding to the possible parents in $\bar{\mathcal{P}}_k$, denote by $\mathbf{B}_I$;

10         Compute $\tilde{a}_{ki}$ by solving an overdetermined linear system, using the non-zero columns of
          $\mathbf{B}_I$ and the corresponding entries in $W_k$; Set the selected entries of $W_k$ as 0 ;

11     $\mathbf{B}[k,:] = W_k$;

12 $\mathbf{A} = \mathbf{I} - \tilde{\mathbf{A}}^{-1}$;

  **Output:** Repermuted matrices $\mathbf{A}$, $\mathbf{B}$ according to the reversed order from Step 1.

---

Condition 1 restricts certain exogenous connections from sources to observed variables in the ground-truth structure. The first part of Condition 1 states that if a source $s$ is a unique component of any possible parent of $x_k$, then $s$ should not have an exogenous connection to $x_k$. The second part states that any possible parent of $x_k$ with no unique components should not overlap in any exogenous connections with $x_k$.[4] In general, $x_k$ can only be exogenously connected to (i) source components that are shared only among its possible parents that have unique component(s), (ii) new source components that are not connected to any of its possible parents. Please refer to Example 7 in Appendix D.5 to see why this condition is necessary for unique identifiability.

### 3.2. The marriage condition

The second part of the necessary and sufficient conditions for unique identifiability of a linear P-SCM is identical to the marriage condition of Hall's marriage theorem (Cameron, 1994). The combinatorial formulation of this theorem considers a collection of finite sets and provides a necessary and sufficient condition (the marriage condition) for the feasibility of selecting a distinct element from each set.

**Lemma 4 (Hall's marriage theorem and marriage condition)** *Let $X$ be a collection of finite subsets of a set $S$. The members of $X$ can be counted with multiplicity (two or more members are identical). Define the mapping $f(x) : X \mapsto S$ s.t. $f$ selects a distinct element from each set $x \in X$. The collection $X$ satisfies the marriage condition if for every sub-collection $C \subseteq X$, $|C| \leq |\bigcup_{x \in C} x|$, i.e., every $C \subseteq X$ covers at least $|C|$ different elements of $S$. Hall's Marriage Theorem states that, the mapping $f$ exists if and only if $X$ satisfies the marriage condition.*

---

4. When an observed variable has multiple unique components, we only need one unique component not to be exogenously connected to $x_k$ to infer the total causal effect. However, there is no prior information about which of these unique components is not exogenously connected to $x_k$, hence none of the unique components can be exogenously connected to $x_k$. For more details, see the necessity proof in Appendix E.3.

In our framework, the collection $X$ corresponds to the family of component sets of possible parents of a variable $x_k$. Specifically, members of the collection $X$ are the finite subsets $\text{Comp}(x_i) \subseteq \mathcal{S}$, $x_i \in \mathcal{P}_k$, $\mathcal{S}$ is the set of sources. The marriage condition translates to the following:

**Condition 2 (The marriage condition)** *For each observed variable $x_k$ with possible parent set $\mathcal{P}_k$, every subset of the possible parent set, $X_C \subseteq \mathcal{P}_k$, satisfies $|X_C| \leq \left| \bigcup_{x_i \in X_C} \text{Comp}(x_i) \right|$. That is, every subset $X_C \subseteq \mathcal{P}_k$ includes at least $|X_C|$ different source variables.*

Condition 2 implies that the possible parent set of every observed variable must have a sufficient number of exogenous connections (from source variables) so that it allows for unique identifiability of the causal effects to this observed variable. Since each variable $x_k$ can be written as a linear combination of the source mixtures in its possible parent set ($\tilde{\mathcal{S}}_k = \{\tilde{s}_i : x_i \in \mathcal{P}_k\}$) plus $\tilde{s}_k$, Condition 2 can be equivalently imposed on the collection $\tilde{\mathcal{S}}_k$: For $\tilde{S}_C \subseteq \tilde{\mathcal{S}}_k$, $|\tilde{S}_C| \leq \left| \bigcup_{\tilde{s}_i \in \tilde{S}_C} \text{Comp}(\tilde{s}_i) \right|$. See Appendix F.3 for the proof. Recall that Condition 1 is also imposed on $\tilde{\mathcal{S}}_k$ (the unique components are defined over the source mixtures in $\mathcal{P}_k$). Please refer to Example 8 in Appendix D.5 to see the necessity of the marriage condition for unique identifiability of a linear P-SCM.

### 3.3. Main result

Our main result states the necessary and sufficient conditions for unique identifiability of the total causal effects among the observed variables and the exogenous connections from the sources.

**Theorem 5** *Let $\mathcal{A} = \{a_{ij}, b_{ik} : i, j \in \{1, \cdots, p\}, k \in \{1, \cdots, m\}\}$. $\mathcal{A}$ fully parameterizes a linear P-SCM. Let $\pi$ denote the Lebesgue measure over $\mathcal{A}$. The causal effects and exogenous connections of a linear P-SCM are uniquely identifiable from observations almost surely (w.r.t. the measure $\pi$) if and only if, for any observed variable $x_k$, both Conditions 1 and 2 are satisfied.*

The proof of Theorem 5 appears in Appendix E. Necessity of the conditions is proved using similar reasoning as in Examples 7 and 8: If either condition is not satisfied, it is impossible to distinguish the true generating model from certain different generating model(s). For sufficiency of the conditions, we propose the P-SCM Recovery algorithm (Algorithm 1) and prove its correctness, i.e., the algorithm recovers the true causal structure when the conditions are satisfied. For each observed variable $x_k$ with possible parent set $\mathcal{P}_k$, Algorithm 1 computes the total causal effects from $x_i \in \mathcal{P}_k$ to $x_k$ using the unique components in $U_k(i)$ in the same iterative manner as described in Definition 3, until no more unique components can be found in the last subset $\mathcal{I}_k$. The algorithm computes the total causal effect from all $x_i \in \mathcal{I}_k$ to $x_k$ by solving an overdetermined linear system. After computing total causal effects from all possible parents to $x_k$, the remaining parts in $x_k$ are considered as exogenous connections.

### 4. Comparison with linear DS-SCM

There are two main differences between our proposed linear P-SCM and linear DS-SCM. The first difference regards the type of latent confounders they allow. From (1), in linear P-SCM, the source mixture associated with each observed variable is comprised of a linear combination of jointly independent sources, which can lead to *linear* latent confounding. That is, the dependency between two source mixtures is due to the fact that they are both linear combinations of the same set of sources. This is not necessarily the case for linear DS-SCM, i.e., the source mixtures can be nonlinear combinations of the sources. The second difference between P-SCM and DS-SCM regards the restriction of distinct sources. Linear P-SCM relaxes the distinct source assumption to

the requirement in Assumption 2 which assumes that each observed variable contains strictly more sources than any of its direct causes. This means that unlike DS-SCM, in our model, an observed variable can be a deterministic function of its direct causes and the latent confounders.

Note that all the works in the literature on linear DS-SCM that we are aware of– with the exception of (Wang and Drton, 2020)– in fact consider an intersection of P-SCM and DS-SCM, in which all three assumptions of linear latent confounding, jointly independent sources (not necessarily independent source mixtures), and Assumption 1 hold (Hoyer et al., 2008; Entner and Hoyer, 2010; Tashiro et al., 2014; Salehkaleybar et al., 2020). In this case, the subclass of P-SCMs representing this subclass are those with matrix $\mathbf{B}$ in (2) of the form $\mathbf{B} = [\mathbf{I} \ \mathbf{B}_s]$. We refer to this subclass of linear G-SCM as linear DS-P-SCM. Therefore, our work strictly expands the considered model space compared to those works. To demonstrate the practical significance of linear P-SCM over linear DS-P-SCM in causal discovery, we show that there are models that can be correctly explained by a linear P-SCM, while linear DS-P-SCMs either are inapplicable or give misleading results (hence the model cannot be recovered using causal discovery methods for linear DS-P-SCMs). These examples appear in Appendix D.2.

### 4.1. Reduction of the conditions for linear DS-P-SCM

To show how our conditions for unique identifiability of linear P-SCMs can be reduced for linear DS-P-SCMs, consider the setting in which each observed variable is associated with a distinct source. Under this setting, the model can be equivalently written as a linear DS-SCM with linear latent confounding and jointly independent sources (see Appendix F.1 for more details). This setting reduces Conditions 1 and 2 in Theorem 5 as follows. First, all possible parents of an observed variable $x_k$ have unique components, i.e., $U_k(i) \neq \emptyset$ for all $x_i \in \mathcal{P}_k$. This further implies that the marriage condition is automatically satisfied. Second, for each $x_k$, the possible parent set $\mathcal{P}_k$ is equivalent to the ancestor set of $x_k$. This follows because, for each $x_i \in \mathcal{P}_k$ with a distinct source $s_j$, since $s_j$ is not connected to any other observed variables including $x_k$, the $s_j$ component in $x_k$ must result from $x_i$. Hence, there must be a causal connection/path from $x_i$ to $x_k$. Theorem 6 below provides the equivalent necessary and sufficient conditions for this setting. See Appendix F.4 for the proof, as well as a graphical representation of the condition.

**Theorem 6** *Suppose every observed variable in the linear P-SCM is associated with a distinct source. The causal effects and exogenous connections of a linear P-SCM are uniquely identifiable from observations almost surely (in the sense described in Theorem 5) if and only if the following condition holds: For a source $s_j$ that is exogenously connected to an observed variable $x_k$, either there are no causal paths from any observed variable containing $s_j$ to $x_k$, or there are at least two distinct causal paths from the observed variable(s) with $s_j$ in their source mixtures to $x_k$.*

**Remark 7** *The condition in Theorem 6 for a linear P-SCM with distinct source implies the condition for unique identifiability of a linear DS-P-SCM in (Salehkaleybar et al., 2020, Theorem 16), but the reverse is not true. Thus, a linear DS-P-SCM can be uniquely identifiable in (Salehkaleybar et al., 2020) but not uniquely identifiable (as a linear P-SCM) in our setting. The reason is that the search space in Theorem 6, which is the whole class of linear P-SCMs, is strictly larger than the search space in (Salehkaleybar et al., 2020), which is linear DS-P-SCMs. See Appendix F.5 for the proof.*

### 4.2. Choice of the representation of linear P-SCM

Note that equivalently we can say P-SCM also requires distinct sources but those sources can have zero variance (i.e., zero). In this case, as shown in Appendix F.2, instead of the form in (2), our model can be equivalently written as

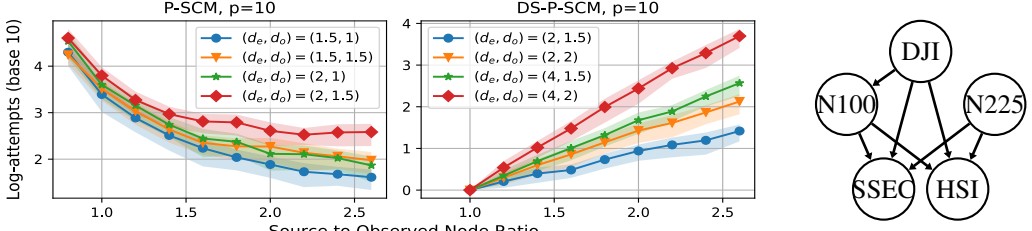

Figure 2: (Left) Satisfiability of our conditions on models generated according to a linear P-SCM or DS-P-SCM. (Right) Recovered causal graphs among five world stock indices.

$$X = \mathbf{A}_{ol}X_l + \mathbf{A}X + S_d, \tag{4}$$

where $X_l$ is the vector of latent confounders; $S_d$ is the vector of (possibly zero) jointly independent distinct sources; $\mathbf{A}_{ol}$ represents the latent confounding. This representation is perhaps more familiar to the reader (as it is considered in most works on linear DS-P-SCM), yet we claim that the representation in (2) is more suitable for the task of identification when zero variance sources exist in the model: First, in order to fully learn the structure of the model in (4), the number of latent confounders must be known in advance (Hoyer et al., 2008; Salehkaleybar et al., 2020), which cannot be deduced from the mixing matrix $\mathbf{W}$ when deterministic relations are present. In this case, we can only learn the sum of the number of latent confounders (i.e., cardinality of $X_l$) and non-zero distinct sources (i.e., support of $S_d$). On the contrary, by considering both latent confounders and non-zero distinct sources as $S$ in (2), we do not need to know the number of latent confounders or the number of non-zero distinct sources in the recovery. Second, our conditions for identifiability of linear P-SCM depend on the exogenous connection matrix $\mathbf{B}$, which includes both latent confounding and non-deterministic relationships in (4) (i.e., matrix $\mathbf{A}_{ol}$ and the non-zero entries in $S_d$). As a result, the identifiability results derived in this work would be more complicated if translated on the model in (4), and we are not aware of a straightforward way to do this translation.

## 5. Numerical experiments

We first provide estimates for the likelihood of satisfiability of the conditions in Theorem 5. Specifically, we randomly generate linear P-SCMs and linear DS-P-SCMs with fixed expected degrees among observed variables and sources, and investigate what portion of them satisfy the conditions. We refer to the generating model as a linear DS-P-SCM when each observed variable in the generated P-SCM has a distinct source. Next, we evaluate the performance of Algorithm 1 for P-SCM recovery on synthetic models under different settings. Finally, for real data, we consider the closing prices of five world stock indices, and model the corresponding returns as a linear P-SCM.[5]

**Satisfiability of the conditions.** We test the satisfiability of our derived conditions on linear P-SCM and linear DS-P-SCM with different pairs of $(d_e, d_o)$, where $d_e(d_o)$ is the expected number of causal (exogenous) connections for each observed (source) variable. Figure 2 (left) shows the average number of generated models (attempts) required to obtain a model that satisfies our conditions, plus/minus half the standard deviations w.r.t. the source to observed node ratio (the number of sources divided by the number of observed variables). The result shows that our conditions are likely to be satisfied on (1) linear P-SCMs when the number of sources is large and the average degrees are small, and (2) linear DS-P-SCMs when both the number of sources and average degrees are small. We

---

5. Our code is available at: https://github.com/Yuqin-Yang/Propagation-SCM.

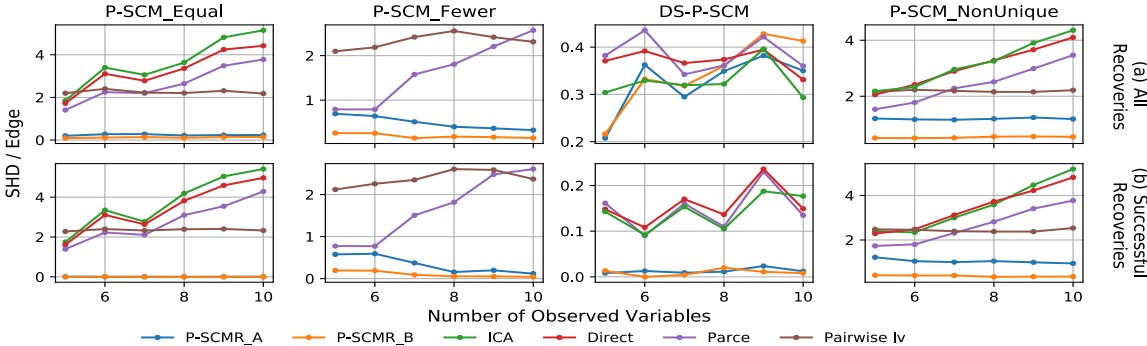

Figure 3: Performance of algorithms in terms of SHD/Edge metric (lower value means better performance). P-SCMR_A, P-SCMR_B represent the recoveries of **A**, **B** by our algorithm.

believe that satisfiability of our conditions depends on (1) the existence and similarity of exogenous connections among observed variables, (2) the ratio of overlap among exogenous connections (how many sources are shared among observed variables) and (3) the structural complexity of the causal connections among observed variables. See Appendix G.1 for more details.

**Performance of recovery.** We evaluate the performance of our recovery algorithm on models generated under each of the following settings: (1) P-SCM with equal number of observed variables and sources (`P-SCM_Equal`); (2) P-SCM with fewer sources than observed variables (`P-SCM_Fewer`) (this indicates that deterministic relation must exist); (3) DS-P-SCM with more sources than observed variables (`DS-P-SCM`); (4) Same as Setting (1) but the model does not satisfy our conditions (`P-SCM_NonUnique`). We only select the generating models which satisfy Conditions 1 and 2 in the first three settings. The sources are independently drawn from uniform distributions; we use FastICA (Hyvarinen, 1999) and ReconstructionICA (Le et al., 2011) for BSS.

We compare our proposed algorithm with ICA-LiNGAM (Shimizu et al., 2006), DirectLiNGAM (Shimizu et al., 2011), ParceLiNGAM (Tashiro et al., 2014), Pairwise lvLiNGAM (Entner and Hoyer, 2010), in terms of the differences between recovered and true adjacency matrix **A**, and recovered and true exogenous connection matrix **B** (after normalization and repermutation).[6] Evaluating **B** is only applicable to our algorithm since other methods do not return **B**. We report the results for the normalized Structural Hamming Distance over true edges (SHD/Edges); comparisons with additional metrics appear in Appendix G.2. We compare the performance under (i) all ICA recoveries and (ii) only the successful recoveries ($\tilde{\mathbf{W}}$ recovered by ICA is highly accurate). Figure 3 shows the performance of recoveries for all algorithms (whenever applicable), averaged over models randomly generated under Settings (1)-(4). The number of observed nodes ranges from 5 to 10.

We observe that when the generating model is a P-SCM, our algorithm can learn the model more accurately than existing algorithms, even if the conditions are not satisfied. This is because existing algorithms may misinterpret the shared sources among observed variables as confounding or direct causal connections, which results in adding additional edges to the recovered graph. When the generating model is a linear DS-P-SCM, the performance of our method is also comparable to existing algorithms. Besides, our algorithm performs significantly better than others when the ICA

---

6. A well-known class of methods for causal structure learning in the presence of latent confounders is FCI algorithm (Spirtes et al., 2000) and its variants. Yet, due to the non-compatibility of the requirements of these methods with our model, we did not compare our approach with these methods (see Appendix G.2 for more details).

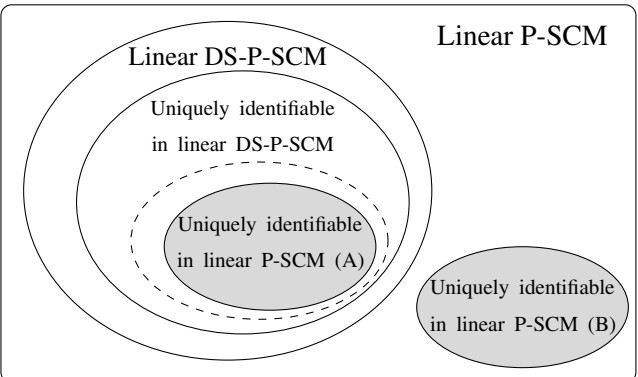

Figure 4: Theoretical results derived in this paper. Dashed circle represent the condition in (Salehka-leybar et al., 2020). Theorem 5 translates to: A linear P-SCM $\mathcal{G} \in A \cup B \Leftrightarrow \mathcal{G}$ satisfies Conditions 1 and 2. Theorem 6 translates to: A linear DS-P-SCM $\mathcal{G} \in A \Leftrightarrow \mathcal{G}$ satisfies the condition in Theorem 6.

recovery is successful. Our algorithm assumes that the true mixing matrix $\mathbf{W}$ can be recovered up to permutation and scaling of its columns, hence depends heavily on the accuracy of BSS recovery.

**Performance on real data.** We test the performance of our recovery algorithm on the daily closing prices of the following five world stock indices: DJI, N225, N100, HSI, SSEC (Hyvärinen et al., 2010; Salehkaleybar et al., 2020). We consider the corresponding return (relative daily change) of each index as an observed variable, and model these observed variables to be generated by a linear P-SCM with five sources, where the sources are non-Gaussian random variables. Using P-SCM Recovery algorithm, the directed graph among these observed variables can be recovered as the right plot in Figure 2. We observe that DJI is a root node of this DAG. Further, DJI, N225 N100 all have causal effects on HSI. Both observations are known to be true either from common belief in economy, or from previous results in (Hyvärinen et al., 2010).

## 6. Conclusion

We proposed the linear propagation SCM (P-SCM), which establishes a way of allowing observed variables in a linear structural causal model to deterministically depend on other observed variables or latent confounders. We considered the problem of finding the necessary and sufficient conditions to uniquely identify a linear P-SCM based on observational data. All theoretical results are summarized in Figure 4. We proposed an algorithm to recover the underlying causal model when a linear P-SCM is assumed to be the data generating process. As mentioned earlier, the performance of our proposed algorithm relies on the accuracy of blind source separation (BSS) recovery. Future directions include investigating algorithms that are robust to BSS errors. Further, it is of interest to investigate whether relaxing the unique identifiability requirement could result in simpler necessary and sufficient conditions that can be efficiently verified.

## Acknowledgments

This research was in part supported by the Swiss National Science Foundation under NCCR Automation, grant agreement 51NF40_180545 and Swiss SNF project 200021_204355 /1.

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

## Appendix A. Related work: A review of causal discovery methods

Conventional methods for recovering causal directions from observational data often make use of Bayesian graphical models (Pearl, 2009; Spirtes et al., 2000). Bayesian networks allow for the use of efficient algorithms, such as constraint-based PC and Fast Causal Inference (FCI) algorithms (Pearl, 2009; Spirtes et al., 2000), and score-based Greedy Equivalent Search (GES) algorithm (Chickering, 2002). However, these methods suffer from an identifiability problem. In particular, they only allow for estimating the underlying causal structure from data up to its *Markov equivalence class*, i.e., the set of structures representing the same conditional independence assertions under causal sufficiency. It turns out the Markov equivalence of the causal structure is a fundamental limit of identification for an underlying data generating process that is a linear DS-SCM with additive Gaussian noises.

Further assumptions on the data generating process can narrow down the set of possible models compatible with the observed data. These assumptions can be made on (i) the functional form relating the observed variables and/or (ii) the distributions of the sources (noises). For example, Shimizu et al. (2006) showed that assuming a linear non-Gaussian additive model (LiNGAM), in which the noises have arbitrary non-Gaussian distributions, enables unique identifiability of the causal structure under causal sufficiency. Alternative assumptions may also lead to unique identifiability: (Peters and Bühlmann, 2014) showed that if all variances of the noises in a linear Gaussian DS-SCM are equal, then one can uniquely identify the causal structure. Hoyer et al. (2009) assumed (i) specific types of nonlinear relationships among observed variables and (ii) additive exogenous noises, and showed that, for the case of two observed variables, nonlinearity allows for identifying the causal structure.

These developments, including LiNGAM algorithm (Shimizu et al., 2006) and its variant DirectLiNGAM (Shimizu et al., 2011) assume causal sufficiency. Without considering the effect of latent confounders, one may infer wrong causal relations among observed variables. Studying causal models with latent confounders might be essential to infer the correct causal structure. Hoyer et al. (2008) and Salehkaleybar et al. (2020) considered an extension of LiNGAM in the presence of latent confounders (lvLiNGAM). Please refer to Appendix C.1 for detailed discussion about both methods.

LiNGAM algorithm is partly based on independent component analysis (ICA) (Comon, 1994; Hyvarinen et al., 2002). The identifiability proof and the recovery algorithm both begin with using ICA to estimate the source mixing matrix from observations. In lvLiNGAM, the number of sources is larger than the number of observed variables since the latent confounders are associated with additional sources. This results in an overcomplete basis for the mixing matrix, and hence lvLiNGAM is based on overcomplete ICA (Lewicki and Sejnowski, 2000; Eriksson and Koivunen, 2004).

Our linear P-SCM generalizes lvLiNGAM to the case when the number of sources can be equal or less than the number of observed variables, since each observed variable is not necessarily associated with a distinct source. Instead, each observed variable is exogenously connected to a subset of the jointly independent set of sources, which allows the observed variable to deterministically depend on other observed variables and latent confounders. We further assume *separability* of the sources from observations, which means all sources composing each observed variable can be correctly estimated, up to scaling and permutation of the coefficients.

SCMs with deterministic relations have been previously considered in the literature. In (Geiger et al., 1990; Spirtes et al., 2000), D-separation[7] condition is proposed for graphically determining conditional independencies when deterministic relations are allowed. Yet, it remains unclear whether D-separation condition can capture all conditional independencies induced from the distribution. Further, a few approaches have been proposed for causal discovery. When the system only consists of two variables with one variable deterministically causing the other, Daniušis et al. (2010) and Janzing et al. (2012) showed that the correct causal direction can be learnt if there is no correlation between the density of the cause variable and the slope of the deterministic function (w.r.t. a reference measure). Their analysis however does not hold for linear relations. Janzing et al. (2010) and Chen et al. (2013) considered the deterministic relation between two high-dimensional observed variables, and showed that the correct causal direction can be recovered using the trace of the covariance matrix of the cause variable and the deterministic function. The trace method in (Chen et al., 2013) was extended by (Zeng et al., 2021) to the case of multi-variable causal discovery. However, all these methods assume that there are only deterministic relations among observed variables.

In a system where both deterministic and non-deterministic relations are present, Scheines et al. (1996) considered recovering the reduced model where all deterministic variables are removed. Luo (2006) adapted the conventional constraint-based IC algorithm (Pearl, 2009) and added new rules in the independence tests to detect deterministic relations. Mabrouk et al. (2014) combined constraint-based methods with greedy search (Tsamardinos et al., 2006), where the deterministic relation is detected by calculating the conditional entropy among variables. Lemeire et al. (2012) introduced information equivalence, where two variables are information equivalent (w.r.t. a third variable) if knowing one variable is equivalent to knowing the other, from the viewpoint of the third variable. Information equivalence is thus a generalization of deterministic relations. Lemeire et al. (2012) adapted the PC algorithm with additional tests to detect information equivalence among variables. The aforementioned methods all suffer from the same identifiability problem as in conventional causal discovery methods, where the underlying causal structure can only be estimated up to certain equivalence classes. In fact, majority of these works do not discuss the capability of identifying the underlying structure by their proposed algorithms (i.e., the equivalence class of the recovery result). Further, these methods do not consider the presence of latent confounders in the system.

## Appendix B. Detailed descriptions of model assumptions

### B.1. P-SCM faithfulness assumption

For a G-SCM, faithfulness assumption means that all marginal and conditional independencies in the underlying causal graph are captured (required) by the Markov condition.[8] For linear DS-SCMs, it is often assumed that any variable is marginally dependent on all its descendants (Salehkaleybar et al., 2020). Thus, the faithfulness assumption translates to the linear coefficients not canceling out causal effects among variables. That is, when multiple causal effects exist from one observed variable to another, their combined effect is not exactly equal to zero.

P-SCM faithfulness assumption extends this idea to linear P-SCMs. Specifically, Assumption 3(a) states that when multiple distinct causal paths exist from some observed variables– containing source a $s_j$– to an observed variable $x_i$, their combined effect do not cancel out one another, or

---

7. This is different from the classical d-separation (with small letter d) described in (Pearl, 2009, Section 1.2).

8. The Markov condition implies a Markov factorization of the underlying probability distribution such that each variable is conditionally independent on its non-descendant variables given its parents.

cancel out the exogenous connection from $s_j$ to $x_i$ (i.e., $s_j$ component in $x_i$ is not equal to zero). This is equivalent to the faithfulness assumption in (Salehkaleybar et al., 2020) for linear DS-SCMs, and is less restrictive than the conventional faithfulness assumption for G-SCMs. Assumption 3(b) considers the coefficients of matrix $\mathbf{B}$, and can be explained as: Any source mixture $\tilde{s}_i$ cannot be written as a linear combination of other source mixtures. Assumption 3(b) is to prevent the case when the coefficients of the exogenous connections from a subset of two or more sources (say $\tilde{S}$) to an observed variable are proportional to the coefficients of the exogenous connections from $\tilde{S}$ to another observed variable. For example, suppose $x_1 = b_{11}s_1 + b_{21}s_2$ and $x_2 = b_{12}s_1 + b_{22}s_2$. Assumption 3(b) prevents $(b_{11}, b_{21})$ from being proportional to $(b_{12}, b_{22})$.

Note that when all model coefficients $\{a_{ij}\}$ and $\{b_{ij}\}$ are independently drawn from continuous distributions, Assumptions (a) and (b) are satisfied almost surely (Meek, 1995), hence P-SCM faithfulness is a reasonable assumption.

### B.2. Separability assumption

Separability can be achieved using blind source separation (BSS) methods, which is to separate a set of sources from a set of their mixtures given little to no information about the sources and the mixing process. For example, when the sources are known to be non-Gaussian random variables, the mixing matrix $\mathbf{W}$ can be recovered using Independent Component Analysis (ICA) or overcomplete ICA methods, depending on the number of observed variables and sources (Comon, 1994; Hyvarinen et al., 2002; Lewicki and Sejnowski, 2000; Eriksson and Koivunen, 2004). Separability can also be achieved when the sources are piecewise constant functionals satisfying a set of mild conditions (Behr et al., 2018). We discuss BSS in more details in Appendix C.

To satisfy the separability property, we assume the availability of a large number of data vectors $\{X^{(i)} : i \in [n]\}$, and that each is generated according to the described process, and using the same coefficients $\{a_{ij}\}$ and $\{b_{ij}\}$.

### Appendix C. Blind source separation methods

Blind source separation (BSS) is the problem of separating a set of source signals (sources) from a set of mixed signals (mixtures) given little to no information about the sources and the mixing process (Comon and Jutten, 2010). In the linear setting where the mixtures are linear combinations of the sources, the blindness often refers to not knowing the source realizations nor the mixing weights. Without any further knowledge about the sources, the problem is infeasible, and hence further assumptions on the sources are needed to facilitate source separation. An example of BSS is the independent component analysis framework in which the sources are assumed to be drawn from independent non-Gaussian distributions. Other examples include the statistical blind source separation regression (SBSSR) model in (Behr et al., 2018) and non-negative matrix factorization in (Donoho and Stodden, 2004). In this section, we provide a brief description of these methods, and present how to utilize them as seeds to our causal structure learning framework.

### C.1. Independent component analysis

Independent component analysis (Comon, 1994) is a statistical technique to separate independent, non-Gaussian random variables (sources) from their observed linear mixtures. Specifically, consider

a vector of $p$ observed variables (mixtures) $X = [x_1, \cdots, x_p]^\top$ which is generated as

$$X = \mathbf{W}S, \tag{5}$$

where $S = [s_1, \cdots, s_m]^\top$ are $m$ unknown real-valued independent non-degenerate random variables (sources). $\mathbf{W}$ is a constant $p \times m$ unknown mixing matrix. $(\mathbf{W}, S)$ is called the *representation* of $X$.

If the observed data is invertible mixtures of non-Gaussian components (i.e., the mixing matrix $\mathbf{W}$ is invertible), Comon (1994) showed that the representation of $X$ can be uniquely identified up to scaling and permutation of the columns of $\mathbf{W}$ (given enough data vectors $X$). Further, if $\mathbf{W}$ is of full column rank and at most one source in $S$ is Gaussian, the representation of $X$ is still identifiable up to scaling and permutation indeterminacies (Eriksson and Koivunen, 2004). For detailed explanation about ICA, the reader can refer to (Hyvärinen and Oja, 2000; Comon and Jutten, 2010).

When the number of observed variables is less than the number of sources, Eriksson and Koivunen (2004) showed that if all the $m$ sources are non-Gaussian and the representation of $X$ is irreducible, then $\mathbf{W}$ can be identified up to scaling and permutation of its columns. Irreducibility means that the columns of $\mathbf{W}$ are pairwise linearly independent. If two columns in $\mathbf{W}$ are linearly dependent, with corresponding sources $s_i$ and $s_j$, then it is impossible to to distinguish between the true representation of $X$ and another representation with $m - 1$ sources, where the two linearly dependent columns are merged into a single column.

We now describe LiNGAM algorithm, which uniquely recovers the causal structure of the underlying model when the mixing matrix is invertible and the sources are non-Gaussian (Shimizu et al., 2006). The underlying DS-SCM can be written as

$$X = \mathbf{A}X + S = \mathbf{W}S \tag{6}$$

where $S$ consists of $m = p$ jointly independent non-Gaussian sources, and $\mathbf{W} = (\mathbf{I} - \mathbf{A})^{-1}$.

Note that (6) fits into the standard ICA framework in (5). Thus, $\mathbf{W}$ can be identified up to scaling and permutation of its columns. Without prior knowledge about the scale and ordering of the sources, the following general ICA model for $X$ holds:

$$X = \tilde{\mathbf{W}}S'. \tag{7}$$

$\tilde{\mathbf{W}}$ is the recovered mixing matrix, with scaling and permutation indeterminacies, and is given by

$$\tilde{\mathbf{W}} = \mathbf{W}\mathbf{P}\mathbf{\Gamma}, \tag{8}$$

where $\mathbf{P}$ is a permutation matrix and $\mathbf{\Gamma}$ is a diagonal scaling matrix. $S'$ contains the corresponding set of sources with reordering and rescaling that correspond to $\mathbf{P}$ and $\mathbf{\Gamma}$.

The corresponding causal model, represented by matrix $\mathbf{A}$, can be uniquely identified due to the acyclicity assumption. Specifically, by some scaling and permutation of its columns and rows, $\tilde{\mathbf{W}}^{-1}$ can be converted uniquely to a lower triangular matrix with all ones on its main diagonal.

In the presence of latent variables, Hoyer et al. (2008) and Salehkaleybar et al. (2020) utilized overcomplete ICA to extend LiNGAM algorithm. Given the output of the overcomplete ICA, the recovered mixing matrix $\tilde{\mathbf{W}}$ is unique up to scaling and permutation of its columns, under the assumption of irreducibility. The number of latent variables in the system is assumed to be known. The task is to identify the columns of $\tilde{\mathbf{W}}$ which correspond to exogenous noises associated with

observed variables, and subsequently apply LiNGAM algorithm after eliminating the columns corresponding to latent variables. Hoyer et al. (2008) proposed an algorithm which brute-forces all possible classifications of the columns of $\tilde{\mathbf{W}}$ (whether each of the columns corresponds to distinct source associated with an observed variable, or to a latent confounder). The algorithm then identifies the possible selections that are compatible with faithfulness and acyclicity assumptions. More recently, Salehkaleybar et al. (2020) proposed a more efficient algorithm that identifies the set of columns corresponding to observed variables. The idea is to learn the skeleton of the underlying graph via pairwise comparisons of the observed variables (based on their corresponding rows in $\tilde{\mathbf{W}}$). This is possible due to faithfulness and acyclicity assumptions. Using the learnt skeleton, the algorithm selects only the columns whose non-zero entries are compatible with the observable descendants of each observed variable.

### C.2. Statistical blind source separation regression model

Statistical blind source separation regression (SBSSR) model is a finite alphabet BSS method proposed in (Behr et al., 2018) to separate piecewise constant source functions from their linear mixtures. Suppose we have a set of $m$ source functions $\mathcal{S} = \{s_1, s_2, \cdots, s_m\}$, each of which consists of an array of constant segments (i.e., a step function) on $[0, 1]$. The function values are selected from a known finite alphabet, yet the jump sizes, numbers, and locations of change points for each function are unknown. Specifically, let $\mathcal{U} = \{u_1, u_2, \cdots, u_k\} \subset \mathbb{R}$ be a known finite and totally ordered alphabet, i.e., $u_1 < u_2 < \cdots < u_k$. Each source function belongs to the class

$$\mathcal{S} = \left\{ \sum_{j=0}^{K} \theta_j \mathbf{1}_{[\tau_j, \tau_{j+1})} : \theta_j \in \mathcal{U}, 0 = \tau_0 < \cdots < \tau_K < \tau_{K+1} = 1, K \in \mathbb{N} \right\}.$$

$K$ is the unknown number of changes, which is assumed to be finite. $\theta_1, \cdots, \theta_K$ are the function values selected from the alphabet $\mathcal{U}$ such that $\theta_j \neq \theta_{j+1}$ for $j = 0, 1, \cdots, K$. $\tau_1, \cdots, \tau_K$ are the change points. We observe samples of $p$ linear mixtures $\{x_k(t), k = 1, \cdots, p\}$ at $n$ uniformly selected location points $t_i$ in $[0, 1]$:

$$x_k(t_i) = \sum_{j=1}^{m} w_{kj} s_j(t_i) + \sigma \epsilon_{ki} = W_k^\top S(t_i) + \sigma \epsilon_{ki}; \qquad i = 1, \cdots, n; \; k = 1, \cdots, p. \quad (9)$$

$W_k = [w_{k1}, \cdots, w_{km}]^\top$ is the vector of mixing weights for mixture $x_k$ with

$$W_k \geq 0, \; \sum_{j=1}^{m} w_{kj} = 1, \qquad k = 1, \cdots, p.$$

$S(t_i) = [s_1(t_i), \cdots, s_m(t_i)]^\top$ is the vector of values of the source functions at point $t_i$ for all $i \in [n]$. $\epsilon_{ki}$ is an additive Gaussian noise with zero mean and unit variance, and $\sigma > 0$ is the known noise variance. Behr et al. (2018) showed that, under mild separability conditions among weights and source functions, both the source functions and the mixing weights can be identified up to permutations, including the number of change points, change point locations, and the function values on each segment for all source functions.

Unlike ICA, the mixing matrix $\mathbf{W}$ in SBSSR does not necessarily need to satisfy the irreducibility assumption, due to the prior knowledge about the sources alphabet. In fact, the recovered mixing

matrix $\tilde{\mathbf{W}}$ using SBSSR can include two linearly independent columns, while the method is able to separate the corresponding sources. However, even with the ability to separate such sources, we require to merge them into one combined source, since the two corresponding generating models are observationally equivalent, as we will shortly explain.

## C.3. Non-negative matrix factorization

Non-negative matrix factorization (NMF) is another BSS method to decompose large matrices into the multiplication of low-rank, non-negative smaller matrices (Lee and Seung, 1999). Suppose each observed variable is an $n_1 \times n_2$ matrix $\mathbf{X}_k$ with rank $r_k$. Using NMF, $\mathbf{X}_k$ can be decomposed into

$$\mathbf{X}_k = \mathbf{W}_k \mathbf{F}_k^\top = \sum_{i=1}^{r_k} W_{ki} F_{ki}^\top. \tag{10}$$

$\mathbf{W}_k$ is a non-negative $n_1 \times r_k$ weight matrix. $\mathbf{F}_k$ is a non-negative $n_2 \times r_k$ feature matrix. $W_{ki}, F_{ki}$ are the column vectors of $\mathbf{W}_k, \mathbf{F}_k$ for all $i = 1, \cdots, r_k$. If we consider the feature vectors $F_{ki}$ as sources, then we can write each observed matrix $\mathbf{X}_k$ into a linear combination of sources (here, both sources and weights are column vectors). Donoho and Stodden (2004) showed that under certain conditions on $\mathbf{W}_k$ and $\mathbf{F}_k$, the factorization (10) is unique up to permutation and scaling of the feature column vectors, i.e., the sources.

## C.4. Application to our model

Under different assumptions on the sources and the mixing matrix, we can use the aforementioned BSS methods to estimate the correct representation of the observed variables. In the BSS methods described in Appendices C.1-C.3, the permutation and scalability indeterminacies always exist, except for SBSSR, where we do not have scalability indeterminacy due to the prior information about the sources alphabets and the mixing weights. In general, more prior information about the sources may eliminate the scalability indeterminacy. Another example where the scalability indeterminacy relaxes is when the sources in the ICA framework are all known to have unit variance. In the same spirit, having prior information about the ordering may eliminate the permutation indeterminacy.

The output of BSS is the factorization of $X$ as in (7), where under permutation and scalability indeterminacies, $\tilde{\mathbf{W}}$ is given by (8), and $S'$ is given by

$$S' = \mathbf{\Gamma}^{-1} \mathbf{P}^\top S.$$

$\mathbf{P}$ is the $m \times m$ permutation matrix with $\mathbf{PP}^\top = \mathbf{I}$, and $\mathbf{\Gamma}$ is the diagonal scaling matrix. Recall that in LiNGAM, the mixing matrix $\mathbf{W} = (\mathbf{I} - \mathbf{A})^{-1}$ is lower triangular with all ones on the main diagonal, since $\mathbf{A}$ is strictly lower triangular due to the DAG-ness assumption of the underlying graph. By contrast, in our model, the mixing matrix in (3) does not possess these structural restrictions since $\mathbf{W} = (\mathbf{I} - \mathbf{A})^{-1}\mathbf{B}$, where $\mathbf{B}$ can have an arbitrary structure.

As in LiNGAM, our recovery algorithm is robust to permutation and scaling ambiguities. From (3) and (8), the recovered mixing matrix in our model can be written as

$$\tilde{\mathbf{W}} = \mathbf{W}\mathbf{P}\mathbf{\Gamma} = (\mathbf{I} - \mathbf{A})^{-1}\mathbf{B}\mathbf{P}\mathbf{\Gamma}.$$

By writing $\tilde{\mathbf{B}}$ as

$$\tilde{\mathbf{B}} = \mathbf{B}\mathbf{P}\mathbf{\Gamma}, \tag{11}$$

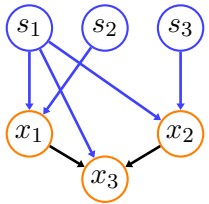

Figure 5: Generating model for Example 1.

we have

$$\tilde{\mathbf{W}} = (\mathbf{I} - \mathbf{A})^{-1}\tilde{\mathbf{B}}. \tag{12}$$

Our recovery algorithm is in fact able to recover $(\mathbf{I} - \mathbf{A})^{-1}$ and $\tilde{\mathbf{B}}$ using $\tilde{\mathbf{W}}$, where $\tilde{\mathbf{B}}$ is a column-repermuted and rescaled version of $\mathbf{B}$. The permutation indeterminacy of $\mathbf{B}$ is immaterial because there exists no ordering among the exogenous connections from the sources, and we are not concerned about the scalability indeterminacy in recovering the strengths of these exogenous connections. Therefore, permutation and scalability indeterminacies do not affect the recovery performance of our algorithm. In the following, we provide an example for the aforementioned indeterminacies.

**Example 1** *Consider the generating model in Figure 5. In the matrix form, the generating model is*

$$\begin{bmatrix} x_1 \\ x_2 \\ x_3 \end{bmatrix} = \begin{bmatrix} 0 & 0 & 0 \\ 0 & 0 & 0 \\ a_{31} & a_{32} & 0 \end{bmatrix} \begin{bmatrix} x_1 \\ x_2 \\ x_3 \end{bmatrix} + \begin{bmatrix} b_{11} & b_{12} & 0 \\ b_{21} & 0 & b_{23} \\ b_{31} & 0 & 0 \end{bmatrix} \begin{bmatrix} s_1 \\ s_2 \\ s_3 \end{bmatrix}. \tag{13}$$

*The observed variables can be represented as linear combinations of the sources as follows:*

$$X = \mathbf{W}S = \begin{bmatrix} b_{11} & b_{12} & 0 \\ b_{21} & 0 & b_{23} \\ w_{31} & a_{31}b_{12} & a_{32}b_{23} \end{bmatrix} \begin{bmatrix} s_1 \\ s_2 \\ s_3 \end{bmatrix},$$

*where $w_{31} = a_{31}b_{11} + a_{32}b_{21} + b_{31}$.*

*Given samples of the observed variables $X$, we can apply BSS methods to learn the representation of $X$ up to permutation and scaling of the columns of $\mathbf{W}$. Suppose the recovered mixing matrix is*

$$\tilde{\mathbf{W}} = \mathbf{W}\mathbf{P}\mathbf{\Gamma} = \begin{bmatrix} \alpha b_{12} & \gamma b_{11} & 0 \\ 0 & \gamma b_{21} & \beta b_{23} \\ \alpha a_{31}b_{12} & \gamma w_{31} & \beta a_{32}b_{23} \end{bmatrix}.$$

*The permutation and scaling matrices are given by*

$$\mathbf{P} = \begin{bmatrix} 0 & 1 & 0 \\ 1 & 0 & 0 \\ 0 & 0 & 1 \end{bmatrix}, \quad \mathbf{\Gamma} = \begin{bmatrix} \alpha & 0 & 0 \\ 0 & \beta & 0 \\ 0 & 0 & \gamma \end{bmatrix},$$

*and the corresponding sources are*

$$S' = \begin{bmatrix} s'_1 \\ s'_2 \\ s'_3 \end{bmatrix} = \begin{bmatrix} \alpha^{-1}s_2 \\ \gamma^{-1}s_1 \\ \beta^{-1}s_3 \end{bmatrix}.$$

$\alpha, \beta, \gamma \neq 0$ *are the scaling coefficients.*

*Given the recovered mixing matrix* $\tilde{\mathbf{W}}$*, our P-SCM recovery algorithm uniquely recovers* $\mathbf{A}$ *and* $\tilde{\mathbf{B}} = \mathbf{B}\mathbf{P}\mathbf{\Gamma}$*, as explained in Section* 3.3*. Thus the recovered model is*

$$X = \mathbf{A}X + \tilde{\mathbf{B}}S' = \begin{bmatrix} 0 & 0 & 0 \\ 0 & 0 & 0 \\ a_{31} & a_{32} & 0 \end{bmatrix} \begin{bmatrix} x_1 \\ x_2 \\ x_3 \end{bmatrix} + \begin{bmatrix} \alpha b_{12} & \gamma b_{11} & 0 \\ 0 & \gamma b_{21} & \beta b_{23} \\ 0 & \gamma b_{31} & 0 \end{bmatrix} \begin{bmatrix} s'_1 \\ s'_2 \\ s'_3 \end{bmatrix}. \tag{14}$$

*Note that* $\tilde{\mathbf{B}}S' = \mathbf{B}S$*, hence the recovered model in* (14) *is identical to the generating model in* (13)*. However, we do not learn the representation* $\mathbf{B}S$ *uniquely, instead we learn* $\tilde{\mathbf{B}}S'$*.*

The irreducibility assumption in our linear P-SCM is without loss of generality. Under P-SCM faithfulness assumption, reducibility occurs if and only if two sources $s_i$ and $s_j$ are exogenously connected to a common observed variable, and are not exogenously connected to any other observed variables. Specifically, we can write the corresponding columns in $\mathbf{W} = (\mathbf{I} - \mathbf{A})^{-1}\mathbf{B}$ (cf. (3)) as

$$[W_i \quad W_j] = (\mathbf{I} - \mathbf{A})^{-1}[B_i \quad B_j],$$

where $B_i$ and $B_j$ are the columns in $\mathbf{B}$ corresponding to sources $s_i$ and $s_j$. Since $(\mathbf{I} - \mathbf{A})^{-1}$ is an invertible square matrix, $W_i$ and $W_j$ are linearly dependent if and only if $B_i$ and $B_j$ are linearly dependent. Under P-SCM faithfulness assumption (b), this happens if and only if $[B_i \quad B_j]$ have only one non-zero row, which means $s_i$ and $s_j$ are exogenously connected to a common observed variable, and to this variable only. Therefore, when $W_i$ and $W_j$ are linearly dependent, i.e., $W_i = \alpha W_j, \alpha \neq 0$, the corresponding sources $s_i, s_j$ appear at each node $x_k$ as $w_{kj}(s_j + \alpha s_i)$, where $w_{kj} \in \mathbb{R}$. Since $s_i$ and $s_j$ are connected to the same variable, replacing the exogenous connections from $s_i, s_j$ by a combined source $s'_i = s_j + \alpha s_i$ in the generating model results in a model that has the same distribution as the original generating model (i.e., an observationally equivalent model).

In conclusion, under different problem settings and assumptions, given observations of variables $X$, we can use BSS methods to recover the mixing matrix $\mathbf{W}$ up to permutation and/or scalability indeterminacies, which have no impact in recovering the true generating model. We call these models separable, as mentioned in Section 2.2.

**Connection to LiNGAM.** According to the above analysis and the comparison between DS-SCM and P-SCM in Section 4, both LiNGAM and lvLiNGAM models are included in our linear P-SCM. Specifically, our model reduces to LiNGAM when each observed variable is directly influenced by a single source, and the sources are jointly independent non-Gaussian random variables with non-zero variances. Here, separability holds by using ICA. Similarly, lvLiNGAM corresponds to the case when the sources are non-Gaussian, and each of the source mixtures $\tilde{s}_i$ has at least one distinct source. As we previously mentioned, for lvLiNGAM, separability hold using overcomplete ICA. In this work, we impose a more general assumption on the *separability* of the source variables from observations. This framework includes ICA and overcomplete ICA as special cases, but is not restricted to these two methods.

## Appendix D. Examples

### D.1. Generalized distinct source assumption

As mentioned in Section 2.1, Assumption 1 is stronger than what is needed for structure learning methods developed for linear DS-SCMs. Here, we provide a weaker version of the distinct source assumption, under which methods developed for linear DS-SCMs still work.

**Assumption 5 (Generalized distinct source assumption)** *A linear G-SCM satisfies the distinct source assumption if for each observed variable $x$, there exists a source such that the information regarding that source in the source mixture of $x$ cannot be fully described by the source mixtures of the rest of the variables.*

The intuition behind Assumption 5 can be explained as follows. If, for example, source $s$ is shared between two variables $x_1$ and $x_2$, yet $x_2$ is only a function of a coarsened version of $s$, then the remaining information of $s$ could be considered as a distinct source for $x_1$. In the following we provide a concrete example where depending on the set of functions $\{g_x\}_{x \in \mathcal{X}}$, the model with the same causal structure may or may not satisfy Assumption 5.

**Example 2** *Consider a linear G-SCM with two observed variables $x_1, x_2$ and two independent source variables $s_1, s_2$. Suppose $s_2$ is discrete with support $\{1, 2, 3, 4\}$, and the generating models of $x_1$ and $x_2$ are*

$$x_1 = g_{x_1}(s_1, s_2) = g_{x_1,1}(s_1) + g_{x_1,2}(s_2),$$
$$x_2 = a_{21}x_1 + g_{x_2}(s_2),$$

*where $g_{x_1,1}$, $g_{x_1,2}$ and $g_{x_2}$ are possibly nonlinear functions.*

- *If all three functions are linear, then the information regarding $s_2$ in the mixture of $x_2$, i.e., $g_{x_2}(s_2)$, can be written as a deterministic function of $g_{x_1,2}(s_2)$, the information regarding $s_2$ in the mixture of $x_1$. That is, $x_2$ does not have a distinct source and hence, the model is not a linear DS-SCM.*

- *If $g_{x_2}(s_2)$ is a linear function of $s_2$, but $g_{x_1,2}(s_2)$ only depends on whether $s_2$ is an even number (e.g., $g_{x_1,2}(s_2) = c_1\mathbf{1}[s_2 \in \{1,3\}] + c_2\mathbf{1}[s_2 \in \{2,4\}]$ for some constants $c_1$, $c_2$); then we can write $s_2$ as the combination of two separate sources $s_3$ and $s_4$, where $s_3, s_4 \in \{0, 1\}$, and*

$$s_2 = 1 \ \ iff \ \ s_4 = 0 \ and \ s_3 = 0;$$
$$s_2 = 2 \ \ iff \ \ s_4 = 1 \ and \ s_3 = 0;$$
$$s_2 = 3 \ \ iff \ \ s_4 = 0 \ and \ s_3 = 1;$$
$$s_2 = 4 \ \ iff \ \ s_4 = 1 \ and \ s_3 = 1.$$

*Note that $s_2 = 2s_3 + s_4 + 1$, and $g_{x_1,2}(s_2)$ is only a function of $s_4$. We can replace $s_2$ in the model by the combination of $s_3$ and $s_4$, and rewrite the model as*

$$x_1 = \tilde{g}_{x_1}(s_1, s_4) = g_{x_1,1}(s_1) + f(s_4),$$
$$x_2 = a_{21}x_1 + g_{x_2}(2s_3 + s_4 + 1),$$

*where $f(s_4) = c_1\mathbf{1}[s_4 = 0] + c_2\mathbf{1}[s_4 = 1]$. Therefore, because of $s_3$, the information regarding $s_2$ in the mixture of $x_2$ (i.e., $g_{x_2}(2s_3 + s_4 + 1)$) cannot be written as a deterministic function*

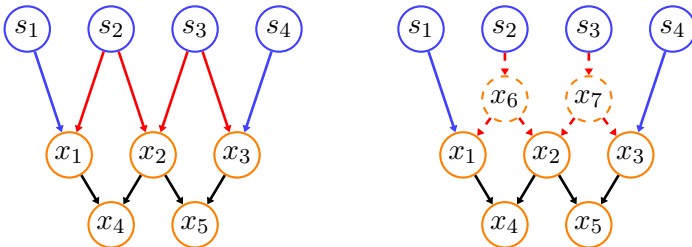

Figure 6: (Left) The ground truth (the true generating model) for Example 3. (Right) The ground truth cannot be modeled as a linear DS-P-SCM with latent confounders. The exogenous connections shared among $x_1$, $x_2$ and $x_3$ in the ground truth are explained here by two additional latent confounders $x_6$ and $x_7$. However, $x_2$, $x_4$, $x_5$ all have no distinct sources, hence deterministically depend on their direct causes and the latent confounder. Red arrows represent the differences.

*of the information regarding $s_2$ in the mixture of $x_1$ (i.e., $f(x_4)$). Further, we can consider $s_3$ as a distinct source of $x_2$, and $x_4$ as a latent confounder between $x_1$ and $x_2$. Therefore, both variables have distinct sources, and the model is a linear DS-SCM.*

## D.2. Examples showing the distinction between linear P-SCM and linear DS-P-SCM

In this subsection, we demonstrate the practical significance of linear P-SCM over a linear DS-P-SCM. Specifically, we answer the following question: Using the more general linear P-SCM, can we correctly identify models that are not identifiable using linear DS-P-SCMs? In the following, we provide two examples where linear P-SCMs can explain the underlying causal model, while linear DS-P-SCMs, with or without latent confounders, either are inapplicable or give misleading results. Therefore, the model cannot be recovered using causal discovery methods for linear DS-P-SCMs, but can be correctly recovered by our P-SCM Recovery algorithm.

**Example 3** *In this example, we begin with the ground truth (i.e., the underlying model), and show that linear DS-P-SCMs fail to explain the model, while our linear P-SCM explains the model. Consider the graph on the left in Figure 6, which represents the true generating model for this example. We have five observed variables $x_1, \cdots, x_5$ and four jointly independent sources $s_1, \cdots, s_4$. Since there are less sources than observed variables, the model cannot be represented by a linear DS-P-SCM with or without latent confounders. This is because each observed variable in linear DS-P-SCMs must be associated with an distinct source that is not shared with other observed variables. Hence the number of independent sources has to be greater than or equal to the number of observed variables in linear DS-P-SCMs.*

*On the other hand, it is easy to see that the ground truth in Figure 6 can be modeled by our linear P-SCM, which allows for observed variables not to have exogenous connections, and explains "latent confounding" simply by "shared" exogenous connections from the sources (for instance, $s_2$ has exogenous connections to both $x_1$ and $x_2$).*

*Lastly, it is worth mentioning that the generating model in this example has a hierarchical structure that entails practical significance: $x_1$, $x_2$ and $x_3$ are exogenously connected to different pairs of sources, while $x_4$ and $x_5$ are composed of different triplets of sources (through different pairs of causal connections from $x_1$, $x_2$ and $x_3$). We thus can treat $\{x_1, x_2, x_3\}$ as one layer, and*

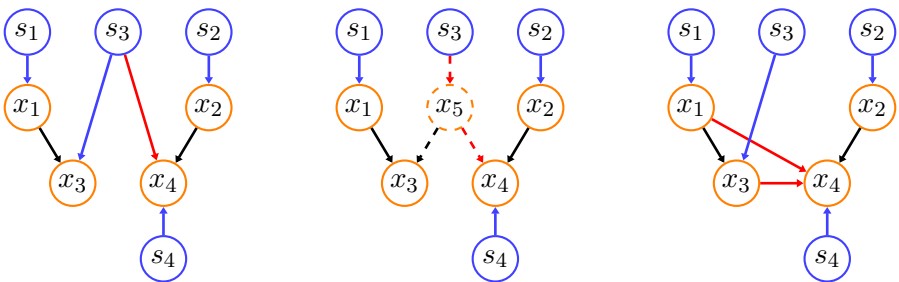

Figure 7: (Left) The ground truth (the true generating model) for Example 4. (Middle) The ground truth can not be modeled as a linear DS-P-SCM with latent confounders: $s_3$ is exogenously connected to two observed variables $x_3$ and $x_4$. A latent confounder $x_5$ is added to the model, which leaves observed variable $x_3$ with no distinct sources. (Right) The ground truth is modeled incorrectly as a linear DS-P-SCM without latent confounders.

$\{x_4, x_5\}$ *as another layer below* $\{x_1, x_2, x_3\}$. *This hierarchical structure may arise naturally in influence propagation problems through networks of nodes. For instance, one layer of twitter users tweet news directly from the news sources (NY-times, Washington-Post, etc), then another layer of users retweet from the first layer, and so on. In conclusion, this example show that our linear P-SCM is more suited to model such a setting than a linear DS-P-SCM.*

**Example 4** *We begin with a ground truth graph and show that– as in the previous example– linear DS-P-SCMs with latent confounders fail to explain the model. Further, we show that linear DS-P-SCMs without latent confounders can explain the model, yet the recovery under this assumption results in misinterpreted edges (low accuracy). A linear P-SCM, on the other hand, explains the model and results in the recovery of a correct network.*

*Consider the graph on the left in Figure 7, which represents the true generating model. We have four observed variables $x_1, \cdots, x_4$ and four jointly independent sources $s_1, \cdots, s_4$. Notice that $s_3$ is exogenously connected to two observed variables. To explain the model by a linear DS-P-SCM, a latent confounder $x_5$ must be added to the model as shown in the middle in Figure 7 (to explain the shared sources), which leaves observed variable $x_3$ having no distinct sources. Therefore the ground truth cannot be explained by a DS-P-SCM with latent confounders. On the contrary, the ground truth can be explained by a linear P-SCM, which allows each observed variable to be exogenously connected to a subset of the sources.*

*Note that, since we have four observed variables and four sources, the model can be modeled by a linear DS-P-SCM without latent confounders (i.e., both models have the same joint distribution over the observed variables), where each observed variable is exogenously connected to one distinct source, and to this source only. This however leads to incorrect recovery as explained in the plot on the right in Figure 7. Specifically, in the true generating model, $s_3$ is exogenously connected to both $x_3$ and $x_4$, while $x_4$ has a distinct source $s_4$. Thus, assuming a linear DS-P-SCM as the generating model, $s_3$ will be the distinct source of $x_3$, and the exogenous connection from $s_3$ to $x_4$ will be interpreted as a causal connection from $x_3$ to $x_4$. This will in turn necessitate the erroneous causal connection from $x_1$ to $x_4$ to cancel out the causal effect of $x_1$ on $x_4$ through $x_3$. Notice here that these additional connections break the marginal independency between $x_1$ and $x_4$, therefore the recovered model is unfaithful to the true generating model.*

## D.3. Example explaining the possible parent set

**Example 5** *Suppose we have 3 observed variables $x_1, x_2, x_3$ and 4 source variables $s_1, s_2, s_3, s_4$. The generating model is given by $x_1 = b_{11}s_1 + b_{12}s_2$; $x_2 = b_{21}s_1 + b_{23}s_3$; $x_3 = a_{31}x_1 + b_{34}s_4$. The observed variables can be written as a linear combination of the sources as follows:*

$$X = \mathbf{W}S = \begin{bmatrix} b_{11} & b_{12} & 0 & 0 \\ b_{21} & 0 & b_{23} & 0 \\ a_{31}b_{11} & a_{31}b_{12} & 0 & b_{43} \end{bmatrix} \begin{bmatrix} s_1 \\ s_2 \\ s_3 \\ s_4 \end{bmatrix}.$$

*The recovered mixing matrix $\tilde{\mathbf{W}}$ is a column-permuted and rescaled version of $\mathbf{W}$. Using $\tilde{\mathbf{W}}$, one possible causal order of the observed variables is $(x_1, x_2, x_3)$, where one may further conclude that $x_1$ and $x_2$ have no parents and $x_3$ has $x_1$ as a parent. Note that both $x_1$ and $x_2$ precede $x_3$ in this causal order, hence both need to be considered when recovering the causal structure of $x_3$.*

*However, the possible parent set of $x_3$ is $\mathcal{P}_3 = \{x_1\}$ (as defined in Section 3): $x_2$ cannot be a parent of $x_3$ since $x_3$ does not have $s_3$ as a component. To conclude, the possible parent set of $x_3$, i.e., $\mathcal{P}_3$, is a strict subset of the observed variable preceding $x_3$ in the casual order deduced from $\tilde{\mathbf{W}}$.*

## D.4. Example explaining the definition of unique component

**Example 6** *Consider a generating model with seven observed variables $x_1, \cdots, x_7$ and eight source variables $s_1, \cdots, s_8$. Let us fix $x_7$. Suppose all the other six variables are possible parents of $x_7$, i.e., $\mathcal{P}_7 = \{x_1, \cdots, x_6\}$. The exogenous connections from the source variables to the observed variables in $\mathcal{P}_7$ are*

$$\tilde{s}_1 = b_{11}s_1 + b_{13}s_3 + b_{14}s_4 + b_{17}s_7;$$
$$\tilde{s}_2 = b_{22}s_2 + b_{23}s_3 + b_{25}s_5;$$
$$\tilde{s}_3 = b_{34}s_4 + b_{36}s_6;$$
$$\tilde{s}_4 = b_{45}s_5 + b_{46}s_6 + b_{48}s_8;$$
$$\tilde{s}_5 = b_{57}s_7 + b_{58}s_8;$$
$$\tilde{s}_6 = b_{67}s_7 + b_{68}s_8.$$

*Let us first consider the possible parent set of $x_7$, i.e., $\mathcal{P}_7$. The index set of $\mathcal{P}_7$ is $J^{(0)} = \{1, \cdots, 6\}$. According to the equations, $s_1$ is connected only to $x_1$, and $s_2$ is connected only to $x_2$. Thus, the unique component set of $x_1$ is $U_7(1) = \{s_1\}$, and the unique component set of $x_2$ is $U_7(2) = \{s_2\}$. The other observed variables in $\mathcal{P}_7$ do not have unique components. Thus, the subset of $\mathcal{P}_7$ with no unique components is $\mathcal{P}_7^{(1)} = \{x_3, x_4, x_5, x_6\}$. The corresponding index set is $J^{(1)} = \{3, 4, 5, 6\}$.*

*Next, we consider only the observed variables in $\mathcal{P}_7^{(1)}$ with indices in $J^{(1)}$. $s_4$ is connected only to $x_3$, and hence $U_7(3) = \{s_4\}$. Similarly, $U_7(4) = \{s_5\}$, and $J^{(2)} = \{5, 6\}$.*

*Finally, each of the observed variables with indices in $J^{(2)}$ ($x_5, x_6$) is connected to both sources $s_7$ and $s_8$. Thus, neither $x_5$ nor $x_6$ has a unique component, and the procedure terminates. To conclude, the set of observed variables with no unique components is $\mathcal{I}_7 = \{x_5, x_6\}$.*

### D.5. Examples showing the necessity of the conditions

In this subsection, we provide examples on how Conditions 1 and 2 are necessary for the unique identifiability of a linear P-SCM. More specifically, in Example 7 below, we show that, for a given observed variable $x_k$, if any source variable that is a unique component of a possible parent $x_i \in \mathcal{P}_k$ has an exogenous connection to $x_k$, then we can not uniquely identify the total causal effect from $x_i$ to $x_k$ in the corresponding generating model. In other words, for unique identifiability of the causal effects to $x_k$, $\tilde{s}_k$ must not contain a unique component of any $x_i \in \mathcal{P}_k$.

Notice that since we consider a fixed observed variable, $x_k$, and aim to recover/estimate the total causal effects from the observed variables in $\mathcal{P}_k$, we do not consider the causal connections among these possible parents.

**Example 7** *Consider the following generating model:*

$$\begin{bmatrix} x_1 \\ x_2 \\ x_3 \end{bmatrix} = \begin{bmatrix} 0 & 0 & 0 \\ 0 & 0 & 0 \\ a_{31} & a_{32} & 0 \end{bmatrix} \begin{bmatrix} x_1 \\ x_2 \\ x_3 \end{bmatrix} + \begin{bmatrix} b_{11} & b_{12} & 0 \\ b_{21} & 0 & b_{23} \\ 0 & b_{32} & 0 \end{bmatrix} \begin{bmatrix} s_1 \\ s_2 \\ s_3 \end{bmatrix}. \tag{15}$$

*The observed variables can be represented as linear combinations of the sources as follows:*

$$X = \mathbf{W}S = \begin{bmatrix} b_{11} & b_{12} & 0 \\ b_{21} & 0 & b_{23} \\ a_{31}b_{11} & a_{31}b_{12} + b_{32} & a_{32}b_{23} \end{bmatrix} \begin{bmatrix} s_1 \\ s_2 \\ s_3 \end{bmatrix}. \tag{16}$$

*Given the observations $X = (x_1, x_2, x_3)^\top$, the mixing matrix $\mathbf{W}$ in (16) can be recovered up to permutation and scaling indeterminacies as described in Appendix C.4. For simplicity, we assume that the recovered mixing matrix $\tilde{\mathbf{W}}$ is the same as $\mathbf{W}$. The intricacies resulting from permutation and scaling indeterminacies is resolved in our proposed P-SCM Recovery algorithm.*

*We assume that the causal structure of $x_1$ and $x_2$ have been correctly recovered as in (15), and consider observed variable $x_3$. From the blind source separation output, i.e., the recovered mixing matrix $\mathbf{W}$ in (16), we deduce that $x_1$ and $x_2$ are possible parents of $x_3$.*

*We can rewrite the generating model of $x_3$ in (15) as follows:*

$$x_3 = a_{31}x_1 + a_{32}x_2 + \tilde{s}_3, \quad \text{where } \tilde{s}_3 = b_{32}s_2. \tag{17}$$

*Notice that $s_2$ is a unique component of $x_1$. The $s_2$ component in $x_3$ (cf. (16)) consists of two parts: $a_{31}b_{12}s_2$ which results from a causal connection from $x_1$, and $b_{32}s_2$ which results from an exogenous connection from $s_2$. In order to learn the causal structure of $x_3$, we ought to be able to differentiate between these two parts from their mixture in the mixing matrix $\mathbf{W}$, which is not possible in this example. Specifically, from the recovered mixing matrix $\tilde{\mathbf{W}} = \mathbf{W}$, we can not distinguish the true generating model in (17) from the two following generating models:*

$$\textbf{Model 1}: \quad x_3^{(1)} = \left(a_{31} + \frac{b_{32}}{b_{12}}\right)x_1 + a_{32}x_2 + \tilde{s}_3^{(1)}, \quad \text{where } \tilde{s}_3^{(1)} = -\frac{b_{11}b_{32}}{b_{12}}s_1,$$

*which corresponds to the $s_2$ component in $x_3$ being caused only by a causal connection from $x_1$; and*

$$\textbf{Model 2}: \quad x_3^{(2)} = a_{32}x_2 + \tilde{s}_3^{(2)}, \quad \text{where } \tilde{s}_3^{(2)} = a_{31}b_{11}s_1 + (a_{31}b_{12} + b_{32})s_2.$$

*which corresponds to the $s_2$ component being caused only by an exogenous connection from $s_2$.*

*Similarly, if the generating model in (17) is such that $\tilde{s}_3 = b_{33}s_3$, the causal effects cannot be uniquely identified, since $s_3$ is a unique component for $x_2$. In conclusion, to uniquely recover the causal structure of $x_3$, neither $s_2$ nor $s_3$ can be included in $\tilde{s}_3$.*

Next, we present an example about the necessity of the marriage condition for unique identifiability a in linear P-SCM. Recall that the marriage condition is: For a given observed variable $x_k$ and its possible parent set $\mathcal{P}_k$, every subset of the possible parents $X_C \subseteq \mathcal{P}_k$ must include at least $|X_C|$ different source variables.

**Example 8 (Marriage condition)** *Consider the following generating model:*

$$
\begin{aligned}
x_1 &= \tilde{s}_1 = b_{11}s_1 + b_{12}s_2; & x_2 &= \tilde{s}_2 = b_{21}s_1 + b_{23}s_3; \\
x_3 &= \tilde{s}_3 = b_{32}s_2 + b_{33}s_3; & x_4 &= \tilde{s}_4 = b_{42}s_2 + b_{44}s_4; \\
x_5 &= \tilde{s}_5 = b_{53}s_3 + b_{54}s_4; & x_6 &= a_{62}x_2 + a_{64}x_4 + a_{65}x_5 + \tilde{s}_6, \quad \tilde{s}_6 = b_{65}s_5.
\end{aligned}
\tag{18}
$$

*The observed variables can be written as follows:*

$$
X = \mathbf{W}S = \begin{bmatrix}
b_{11} & b_{12} & 0 & 0 & 0 \\
b_{21} & 0 & b_{23} & 0 & 0 \\
0 & b_{32} & b_{33} & 0 & 0 \\
0 & b_{42} & 0 & b_{44} & 0 \\
0 & 0 & b_{53} & b_{54} & 0 \\
w_{61} & w_{62} & w_{63} & w_{64} & w_{65}
\end{bmatrix}
\begin{bmatrix}
s_1 \\ s_2 \\ s_3 \\ s_4 \\ s_5
\end{bmatrix}.
\tag{19}
$$

*We recover the mixing matrix $\mathbf{W}$ using BSS. As in the previous example, we assume that $\tilde{\mathbf{W}} = \mathbf{W}$. Consider the observed variable $x_6$. From $\mathbf{W}$, the possible parents of $x_6$ are $\mathcal{P}_6 = \{x_1, \cdots, x_5\}$. Notice that the marriage condition is not satisfied in this example: $\left|\bigcup_{x_i \in \mathcal{P}_6} \mathrm{Comp}(x_i)\right| = 4$ while $|\mathcal{P}_6| = 5$, therefore, $|\mathcal{P}_6| \not\leq \left|\bigcup_{x_i \in \mathcal{P}_6} \mathrm{Comp}(x_i)\right|$. In the following, we show how this renders the unique identifiability to be impossible.*

*Consider $X_{[1:4]} = (x_1, x_2, x_3, x_4)^\top$. These observed variables only contain the source variables $S_{[1:4]} = (s_1, \cdots, s_4)^\top$, and thus can be written as $X_{[1:4]} = \mathbf{W}_0 S_{[1:4]}$, where $\mathbf{W}_0$ is the upper left $4 \times 4$ submatrix of $\mathbf{W}$ in (19). $\mathbf{W}_0$ is invertible due to P-SCM faithfulness assumption. Now, $x_5$ contains only the sources $(s_3, s_4)$, and hence can be written as a linear combination of $X_{[1:4]}$ as follows:*

$$
x_5 = [0\ 0\ b_{53}\ b_{54}]\mathbf{W}_0^{-1}X_{[1:4]} = \sum_{i=1}^{4} c_i x_i,
\tag{20}
$$

*where $c_i, i = 1, \cdots, 4$ can be calculated using the exogenous connections. The fact that $x_5$ is linearly dependant on (i.e., can be represented as a linear combination of) $x_1, \cdots, x_4$ results in impossibility of learning the causal connections to $x_6$ from observations. Specifically, it is impossible to distinguish the true generating model of $x_6$ in (18) from the following model, in which $x_5$ in the true generating model is replaced by (20):*

$$
x_6 = c_1 a_{65} x_1 + (c_2 a_{65} + a_{62})x_2 + c_3 a_{65} x_3 + (c_4 a_{65} + a_{64})x_4 + \tilde{s}_6.
\tag{21}
$$

*Therefore, the generating model is not uniquely identifiable.*

*As we have seen in this example, unique identifiability of causal effects to $x_6$ fails because, in the representation of X recovered by BSS (cf. (19)), $x_5$ is linearly dependent on the remaining possible parents of $x_6$, i.e., $X_{[1:4]}$. In Appendix F.3, we show that linear independence among possible parents in $\mathcal{P}_k$ holds if and only if the marriage condition is satisfied.*

## Appendix E. Proof of main theorem

### E.1. Matrix representation of the necessary and sufficient conditions

Let us first define the existing component set $\mathcal{E}_k$ for each observed variable $x_k$ as follows.

**Definition 8** *The* existing component set *of $x_k$, denoted by $\mathcal{E}_k$, is the set of all components in the possible parents of $x_k$, i.e., $\mathcal{E}_k = \bigcup_{x_i \in \mathcal{P}_k} \text{Comp}(x_i)$.*

Recall that matrix $\mathbf{B}$ in (2) consists of all the exogenous connections from sources to observed variables. Let $\mathbf{B}_k$ denote the submatrix of $\mathbf{B}$ with rows corresponding to the possible parents of $x_k$, i.e., $\mathcal{P}_k$, and columns corresponding to the existing component set of $x_k$, i.e., $\mathcal{E}_k$. That is,

$$\mathbf{B}_k = [\mathbf{B}_{ij}]_{i,j: \, x_i \in \mathcal{P}_k, \, s_j \in \mathcal{E}_k} \tag{22}$$

We repermute the rows and columns of $\mathbf{B}_k$, and label the columns, as follows:

(i) We find the columns in $\mathbf{B}_k$ with only one non-zero entry and label these columns as type-1. We repermute the columns of $\mathbf{B}_k$ such that all type-1 columns are at the leftmost of $\mathbf{B}_k$.

(ii) We repermute the rows of $\mathbf{B}_k$ such that all nonzero entries in type-1 columns appear in the upper rows. Note that two type-1 columns can have their non-zero entry in the same row.

(iii) Among the columns that are not type-1 (have two or more non-zero entries), we find those columns with their non-zero entries appearing in the non-zero part of type-1 columns (i.e., the upper non-zero rows). We label these columns as type-2. We repermute the columns of $\mathbf{B}_k$ such that type-2 columns are next to type-1 columns. The matrix $\mathbf{B}_k$ can be now written as

$$\mathbf{B}_k = \begin{bmatrix} \mathbf{U}_1 & \mathbf{X}_1 & \mathbf{Y}_1 \\ \mathbf{0} & \mathbf{0} & \mathbf{Z}_1 \end{bmatrix}. \tag{23}$$

$[\mathbf{U}_1; \mathbf{0}]$ are the type-1 columns, $[\mathbf{X}_1; \mathbf{0}]$ are the type-2 columns, and $[\mathbf{Y}_1; \mathbf{Z}_1]$ are the remaining columns. Each column of $\mathbf{Z}_1$ has at least one non-zero entry.

(iv) We repeat steps (i) to (iii) to permute the columns and rows of $\mathbf{Z}_1$ such that

$$\mathbf{Z}_1 = \begin{bmatrix} \mathbf{U}_2 & \mathbf{X}_2 & \mathbf{Y}_2 \\ \mathbf{0} & \mathbf{0} & \mathbf{Z}_2 \end{bmatrix}. \tag{24}$$

The columns of $\mathbf{B}_k$ corresponding to the columns of $\mathbf{U}_2$ and $\mathbf{X}_2$ (which were unlabeled in (23)) are now labeled as type-1 and type-2 respectively.

(v) For $\mathbf{Z}_n$, $n = 2, 3, \cdots$, we repeat steps (i) to (iii) and label the corresponding columns in $\mathbf{B}_k$ as in (iv); until this factorization does not hold for $\mathbf{Z}_N$ for some $N \in \mathbb{N}$. We label the columns of $\mathbf{B}_k$ that correspond to the columns of $\mathbf{Z}_N$ as type-3 columns.

The column and row permutations in (i)-(v) are analogous to the iterative process in Definition 3. Type-1 columns in (i) correspond to the unique components in the first iteration, i.e., among possible parents in $\mathcal{P}_k$. The rows selected in (ii) correspond to the possible parents in $\mathcal{P}_k \setminus \mathcal{P}_k^{(1)}$. Type-2 columns correspond to the sources that are shared only among the possible parents in $\mathcal{P}_k \setminus \mathcal{P}_k^{(1)}$. $\mathbf{Z}_1$ in (23) represents the exogenous connections of observed variables in $\mathcal{P}_k^{(1)}$. For the iteration in (i)-(v), $\mathbf{Z}_n$, $n = 1, 2, \cdots, N$, can be written as

$$\mathbf{Z}_n = [\mathbf{B}_{ij}]_{i,j:\, x_i \in \mathcal{P}_k^{(n)},\, s_j \in \bigcup_{i \in J^{(n)}} \mathrm{Comp}(\tilde{s}_i)}.$$

Lastly, the rows in $\mathbf{Z}_N$ correspond to the possible parents in $\mathcal{I}_k$, and the sources shared among these variables correspond to type-3 columns.

Given the matrix permutation and labeling in (i)-(v), the unique components condition translates to: Each observed variable $x_k$ can only be exogenously connected to either source variables corresponding to type-2 columns in $\mathbf{B}_k$, or new source components that are not in $\mathcal{E}_k$ (i.e., do not correspond to columns in $\mathbf{B}_k$). Further, the marriage condition translates to the matrix $\mathbf{B}_k$ being of full row rank, which is equivalent to $\mathbf{Z}_N$ being of full row rank. The equivalence between the marriage condition and the full-row rank of $\mathbf{B}_k$ is explained in Appendix F.3.

### E.2. Proof of sufficiency

For any data generating model that satisfies Conditions 1 and 2 in Sections 3.1 and 3.2, we show that using the recovered mixing matrix $\tilde{\mathbf{W}}$, we can construct an algorithm that uniquely recovers both the causal effects among observed variables and the exogenous connections from the sources. Specifically, we show that P-SCM Recovery Algorithm (Algorithm 1) recovers the exact matrix of total causal effects, i.e., $(\mathbf{I} - \mathbf{A})^{-1}$, and the exogenous connection matrix $\mathbf{B}$ up to permutation and scaling, i.e., $\tilde{\mathbf{B}}$ in (11). The row permutation in Step 1 of Algorithm 1 has the property that for each observed variable $x_k$, the possible parents of $x_k$ are all preceding $x_k$ in the row ordering. This follows because for every $x_i \in \mathcal{P}_k$, $\mathrm{Comp}(x_i) \subsetneq \mathrm{Comp}(x_k)$ according to Definition 2. Without loss of generality, assume that the ordering among observed variables after the permutation in Step 1 is a natural ordering. In this case, we have $\mathcal{P}_k \subseteq \{x_1, \cdots, x_{k-1}\}$ for all $x_k$.

We provide the proof by induction on the index of the observed variables $k$, where the induction base is $k = 1$. For $k = 1$, there are no observed variables in $\mathcal{P}_1$, hence all the source components of $x_1$ must result from exogenous connections, which are learnt from the recovered mixing matrix. For the induction hypothesis, we assume that the total causal effects among $\{x_1, \cdots, x_{k-1}\}$ and their exogenous connections from the sources are given. We show that we can recover the total causal effects as well as the exogenous connections to $x_k$.

From the induction hypothesis, we know the exogenous connections (source mixtures) to the possible parents of $x_k$, i.e., $\tilde{\mathcal{S}}_k = \{\tilde{s}_i : x_i \in \mathcal{P}_k\}$. Our algorithm follows a similar iterative procedure as in Definition 3, in order to identify the unique components of each $x_i \in \mathcal{P}_k$, while simultaneously solve for the total causal effect from $x_i$ to $x_k$.

First, the algorithm finds the observed variables with unique components among all possible parents in $\mathcal{P}_k$, i.e., $x_i \in \mathcal{P}_k \setminus \mathcal{P}_k^{(1)}$ (see Section 3.1). For each $x_i \in \mathcal{P}_k \setminus \mathcal{P}_k^{(1)}$ with a unique component $s_j$, it follows from the unique component condition that $s_j$ is not exogenously connected to $x_k$. Thus, the $s_j$ component in $x_k$ must result from the causal connection/path from $x_i$. The strength of the causal effect can be correctly recovered via dividing $s_j$ component in $x_k$ by $s_j$ component in $\tilde{s}_i$, i.e., $\tilde{w}_{kj}/\tilde{b}_{ij}$. Subsequently, we correctly recover the total causal effect from all $x_i \in \mathcal{P}_k \setminus \mathcal{P}_k^{(1)}$ to $x_k$.

**Remark 9** *Note that $s_j$ component in $x_i$ only results from an exogenous connection from $s_j$ to $x_i$, since the parents of $x_i$ are also in $\mathcal{P}_k$ and $s_j$ is a unique component. Hence the $s_j$ component in $\tilde{s}_i$ is the same as the $s_j$ component in $x_i$. This explains our use of "dividing by $s_j$ coefficient in $x_i$", instead of in $\tilde{s}_i$, in Section 3.1.*

Next, we find the total causal effects from $x_i \in \mathcal{P}_k^{(1)}$. Similarly, consider $x_i \in \mathcal{P}_k^{(1)} \setminus \mathcal{P}_k^{(2)}$ with a unique component $s_j$. It follows from the unique components condition that $s_j$ is not exogenously connected to $x_k$. Thus, $s_j$ component in $x_k$ results from the following effects: (i) causal connection/path from $x_i$ and/or (ii) causal connections/paths from observed variables in $\mathcal{P}_k \setminus \mathcal{P}_k^{(1)}$. The latter causal effects are recovered in the last step and subtracted from $x_k$, and hence we calculate the total causal effect from $x_i$ to $x_k$ by the residual of $s_j$ component in $x_k$ divided by $s_j$ component in $\tilde{s}_i$. Therefore, we correctly recover the total causal effects from all $x_i \in \mathcal{P}_k^{(1)} \setminus \mathcal{P}_k^{(2)}$ to $x_k$.

Repeat the last step for $x_i \in \mathcal{P}_k^{(n)} \setminus \mathcal{P}_k^{(n+1)}$ for $n = 2, \cdots, N-1$, i.e., all observed variables with unique components. Next, we find the total causal effects from $x_i \in \mathcal{I}_k$ (with no unique components) to $x_k$. The unique components condition implies that $x_k$ cannot be exogenously connected to any source variable that belongs to $\bigcup_{i \in J^{(N)}} \mathrm{Comp}(\tilde{s}_i)$ (recall that $J^{(N)}$ is the index set of $\mathcal{I}_k$). That is, the exogenous connections to $x_k$ can not overlap with the exogenous connections to any $x_i \in \mathcal{I}_k$. Then, we recover the total causal effects from $x_i \in \mathcal{I}_k$ to $x_k$ using the residuals of the set of sources $\bigcup_{i \in J^{(N)}} \mathrm{Comp}(\tilde{s}_i)$ in $x_k$ (after all subtractions in the previous steps). This results in an overdetermined linear system of the total causal effects:

$$\mathbf{w} = \mathbf{Z}_N^\top \tilde{\mathbf{a}}_k, \quad \mathbf{Z}_N = [\tilde{\mathbf{B}}_{ij}]_{i,j: \, x_i \in \mathcal{I}_k, \, s_j \in \bigcup_{i \in J^{(N)}} \mathrm{Comp}(\tilde{s}_i)}. \tag{25}$$

$\mathbf{w}$ is the vector of the residuals of the source components in $x_k$ that correspond to $\bigcup_{i \in J^{(N)}} \mathrm{Comp}(\tilde{s}_i)$; $\tilde{\mathbf{a}}_k$ is the vector of total causal effects from the observed variables in $\mathcal{I}_k$ to $x_k$; $\mathbf{Z}_N$ represents the exogenous connections to observed variables in $\mathcal{I}_k$ (see Appendix E.1).

According to the marriage condition, $\mathbf{Z}_N$ is of full row rank, see Appendix E.1. Thus, the overdetermined linear system in (25) has at most one solution which corresponds to the true total causal effects from each $x_i \in \mathcal{I}_k$. Note that $\tilde{\mathbf{B}}_k$ has the same rank as $\mathbf{B}_k$ since permuting the columns does not change the rank. To sum up, Algorithm 1 correctly recovers the total causal effects from the possible parents of $x_k$ ($x_i \in \mathcal{P}_k$) to $x_k$. The remaining source components in $x_k$ represent the exogenous connections from the corresponding sources to $x_k$, which can be recovered up to permutation and scalability indeterminacies. The recovered matrix $\tilde{\mathbf{B}}$ has the same column-permutation and scaling as the output of BSS $\tilde{\mathbf{W}}$. That is, Algorithm 1 is able to recover the exact adjacency matrix $\mathbf{A}$ using the unique components and/or matrix inversion, by shifting all the indeterminacies into the residual values to recover $\tilde{\mathbf{B}}$. Recall, for matrix $\mathbf{B}$, the column permutation indeterminacy is irrelevant; the scalability indeterminacy is unavoidable in our setting unless we have prior information about the sources.

The column permutation is maintained for $\tilde{\mathbf{B}}$ because the order of the sources, given in $\tilde{\mathbf{W}}$, is not changed throughout Algorithm 1. For scalability of the causal effects, suppose the learnt exogenous connections from a source $s_l$ to all $x_i \in \mathcal{P}_k$ (i.e., the column of $\tilde{\mathbf{B}}_k$ corresponding to $s_l$) share the same scale as the corresponding column in $\tilde{\mathbf{W}}$. Algorithm 1 first recovers the correct total causal effect $\tilde{a}_{ki} = \tilde{w}_{kj}/\tilde{b}_{ij}$ (cf. Step 6) using the unique components for $x_i \in \mathcal{P}_k \setminus \mathcal{P}_k^{(1)}$. The residual of $s_l$ in $x_k$ after subtracting the effect of $x_i$ (Step 7), which is $\tilde{w}_{kl} - \tilde{a}_{ki}\tilde{b}_{il}$, shares the same scale as $\tilde{w}_{kl}$. Similarly, for all $x_i$ with unique components, i.e., $x_i \in \mathcal{P}_k \setminus \mathcal{I}_k$, Algorithm 1 recovers the

correct total causal effect $\tilde{a}_{ki}$, and the residual shares the same scale as $\tilde{w}_{kl}$. For $x_i \in \mathcal{I}_k$, Algorithm 1 recovers the correct total causal effect using (25) (cf. Step 10), since each column of $\mathbf{Z}_N$ in (25) shares the same scale as the corresponding entry of $\mathbf{w}$. Finally, the remaining $s_l$ component in $x_k$ shares the same scale as $\tilde{w}_{kl}$, which is then considered as the exogenous connection $\tilde{b}_{kl}$. Therefore, the learnt exogenous connections from $s_l$ to all observed variables $x_k$, which are in the column of $\tilde{\mathbf{B}}$ corresponding to $s_l$, share the same scale as the corresponding column in $\tilde{\mathbf{W}}$.

In conclusion, given the causal structure and exogenous connections to the possible parents of $x_k$, we are able to recover the exact total causal effects to $x_k$ along with its exogenous connections. This establishes the induction step and completes our sufficiency proof.

**Remark 10** *As we mentioned earlier, for each observed variable $x_k$, the possible parents of $x_k$, i.e., $\mathcal{P}_k$, are all preceding $x_k$ in the row ordering deduced in Step 1 of Algorithm 1. Since the ancestors of $x_k$ are also included in $\mathcal{P}_k$, this implies that by sorting the observed variables according to the number of source components in the recovered mixing matrix $\tilde{\mathbf{W}}$, we can find a causal order among the observed variables which is consistent with the correct order. This method is faster than the pairwise comparison in (Salehkaleybar et al., 2020, Lemma 5).*

### E.3. Proof of necessity

We show that for any generating model $\mathcal{G}$ that violates either the unique component condition or the marriage condition, by using the corresponding $\tilde{\mathbf{W}}$, it is not possible to distinguish between the true generating model $\mathcal{G}$ and another generating model $\mathcal{G}'$. Without loss of generality, assume that the observed variables are ordered such that for any $x_k$, $\mathcal{P}_k \subseteq \{x_1, \cdots, x_{k-1}\}$. This ordering can be achieved by permuting the rows of $\tilde{\mathbf{W}}$ to the corresponding order.

Now, consider a generating model $\mathcal{G}$ that does not satisfy either or both of the conditions. Suppose $x_k$ has the smallest index among the observed variables such that the conditions are not satisfied. That is, the submodel $\{x_1, \cdots, x_{k-1}\}$ does satisfy the unique component and marriage conditions. We can write the generating model $\mathcal{G}$ of $x_k$ as:

$$x_k = \sum_{i:x_i \in \mathcal{P}_k} \tilde{a}_{ki} \tilde{s}_i + \tilde{s}_k, \quad \tilde{s}_k = \sum_{s_j \in \mathrm{Comp}(\tilde{s}_k)} b_{kj} s_j. \tag{26}$$

where we redefine $\tilde{a}_{ki}$ to be the total causal effect from observed variable $x_i$ to $x_k$, and $b_{kj}$ is the strength of the exogenous connection from $s_j$ to $x_k$. Non-satisfiability of the theorem conditions can only happen when either of the following cases hold.

(i) $\exists x_{i_0} \in \mathcal{P}_k$ with a non-empty unique component set, such that *at least one unique component* of $x_{i_0}$ is exogenously connected to $x_k$.

(ii) $\exists x_{i_0} \in \mathcal{P}_k$ with an empty unique component set such that *at least one component* of $\tilde{s}_{i_0}$ is exogenously connected to $x_k$.

(iii) The collection of component sets of the source mixtures $\tilde{\mathcal{S}}_k = \{\mathrm{Comp}(\tilde{s}_i) : x_i \in \mathcal{P}_k\}$ does not satisfy the marriage condition.

**Case (i).** Suppose there exists an index $i_0$ such that $U_k(i_0) \cap \mathrm{Comp}(\tilde{s}_k) \neq \emptyset$. Let $s_{j_0} \in U_k(i_0) \cap \mathrm{Comp}(\tilde{s}_k)$, i.e., $s_{j_0}$ is a unique component of $x_{i_0}$ and is also exogenously connected to

$x_k$. In the following, we show we cannot determine if the component $s_{j_0}$ in $x_k$ results from a causal connection/path or an exogenous connection to $x_k$, or both.

We first take out the terms that include $\tilde{s}_{i_0}$ and $s_{j_0}$ from $x_k$ in (26), and rewrite (26) as

$$x_k = \sum_{\substack{i:x_i \in \mathcal{P}_k \\ i \neq i_0}} \tilde{a}_{ki} \tilde{s}_i + \tilde{a}_{ki_0} \tilde{s}_{i_0} + \sum_{\substack{s_j \in \mathrm{Comp}(\tilde{s}_k) \\ j \neq j_0}} b_{kj} s_j + b_{kj_0} s_{j_0}, \quad b_{kj_0} \neq 0. \tag{27}$$

Next, consider the exogenous connections to $x_{i_0}$ (i.e., $\tilde{s}_{i_0}$), which can be expressed as

$$\tilde{s}_{i_0} = \sum_{\substack{s_{j'} \in \mathrm{Comp}(\tilde{s}_{i_0}) \\ j' \neq j_0}} b_{i_0 j'} s_{j'} + b_{i_0 j_0} s_{j_0}, \quad b_{i_0 j_0} \neq 0. \tag{28}$$

By substituting (28) in (27), we get

$$
\begin{aligned}
x_k &= \sum_{\substack{i:x_i \in \mathcal{P}_k \\ i \neq i_0}} \tilde{a}_{ki} \tilde{s}_i + \sum_{\substack{s_j \in \mathrm{Comp}(\tilde{s}_k) \\ j \neq j_0}} b_{kj} s_j + b_{kj_0} s_{j_0} + \tilde{a}_{ki_0} \left( \sum_{\substack{s_{j'} \in \mathrm{Comp}(\tilde{s}_{i_0}) \\ j' \neq j_0}} b_{i_0 j'} s_{j'} + b_{i_0 j_0} s_{j_0} \right) \\
&= \sum_{\substack{i:x_i \in \mathcal{P}_k \\ i \neq i_0}} \tilde{a}_{ki} \tilde{s}_i + \sum_{\substack{s_j \in \mathrm{Comp}(\tilde{s}_k) \\ j \neq j_0}} b_{kj} s_j + \sum_{\substack{s_{j'} \in \mathrm{Comp}(\tilde{s}_{i_0}) \\ j' \neq j_0}} \tilde{a}_{ki_0} b_{i_0 j'} s_{j'} + (\tilde{a}_{ki_0} b_{i_0 j_0} + b_{kj_0}) s_{j_0}. \tag{29}
\end{aligned}
$$

Suppose the total causal effects from all observed variables $x_i \in \mathcal{P}_k \setminus \{x_{i_0}\}$ are recovered as in (29). Now, our task is to recover the total causal effect from $x_{i_0}$ to $x_k$.

Let us consider the $s_{j_0}$ component in (29). It consists of two parts: $\tilde{a}_{ki_0} b_{i_0 j_0} s_{j_0}$ which results from the causal connection/path from $x_{i_0}$, and $b_{kj_0} s_{j_0}$ which results from the exogenous connection from $s_{j_0}$. In order to learn the causal structure, we ought to be able to differentiate between these two parts from their mixture in (29), which is not possible. More specifically, we can not distinguish the true model from the following two generating models:

*Model 1:*

$$x_k^{(1)} = \sum_{\substack{i:x_i \in \mathcal{P}_k \\ i \neq i_0}} \tilde{a}_{ki} \tilde{s}_i + (\tilde{a}_{i_0} + \frac{b_{kj_0}}{b_{i_0 j_0}}) \tilde{s}_{i_0} + \tilde{s}_k^{(1)};$$

$$\tilde{s}_k^{(1)} = \sum_{\substack{s_j \in \mathrm{Comp}(\tilde{s}_k) \\ j \neq j_0}} b_{kj} s_j - \sum_{\substack{s_{j'} \in \mathrm{Comp}(\tilde{s}_{i_0}) \\ j' \neq j_0}} \frac{b_{kj_0} b_{i_0 j'}}{b_{i_0 j_0}} s_{j'}, \tag{30}$$

which corresponds to the $s_{j_0}$ component in $x_k$ being caused only by the causal effect from $x_{i_0}$.

*Model 2:*

$$x_k^{(2)} = \sum_{\substack{i:x_i \in \mathcal{P}_k \\ i \neq i_0}} \tilde{a}_{ki} \tilde{s}_i + \tilde{s}_k^{(2)};$$

$$\tilde{s}_k^{(2)} = \sum_{\substack{s_j \in \mathrm{Comp}(\tilde{s}_k) \\ j \neq j_0}} b_{kj} s_j + \sum_{\substack{s_{j'} \in \mathrm{Comp}(\tilde{s}_{i_0}) \\ j' \neq j_0}} \tilde{a}_{ki_0} b_{i_0 j'} s_{j'} + (\tilde{a}_{ki_0} b_{i_0 j_0} + b_{kj_0}) s_{j_0}, \tag{31}$$

which corresponds to the $s_{j_0}$ component being caused only by the exogenous connection from $s_{j_0}$. In contrast, if the unique component $s_{j_0}$ is not exogenously connected to $x_k$, then we can use this unique component to compute the total causal effect from $x_{i_0}$ to $x_k$ (i.e., $\tilde{a}_{ki_0}$), which in turn can be used to determine if there exist exogenous connections from the remaining components in $\tilde{s}_{i_0}$ to $x_k$.

Notice that an observed variable $x_i$ may have multiple unique components in the unique component set $U_k(i)$. Suppose there exists another source $s_{l_0}$ which is a unique component of $x_{i_0}$ and is not exogenously connected to $x_k$, while the unique component $s_{j_0}$ of $x_{i_0}$ is exogenously connected to $x_k$. Since we have no prior information about which of the two unique components is not exogenously connected to $x_k$, there are two candidate models that we cannot distinguish between: In each of these two models, the total causal effect from $x_{i_0}$ to $x_k$ is calculated using one of the two unique components in $x_{i_0}$, and there is an extra exogenous connection from the other unique component. Thus none of the unique components in $U_k(i)$ can be exogenously connected to $x_k$. Note that having multiple unique components for a single observed variable does not contradict the irreducibility assumption. This is because unique components are defined over the source mixtures of the possible parents of $x_k$, i.e., $\tilde{\mathcal{S}}_k$, and irreducibility assumption is defined over the whole mixing matrix.

**Case (ii).** Suppose there exists an index $i_0$ such that $\mathrm{Comp}(\tilde{s}_{i_0}) \cap \mathrm{Comp}(\tilde{s}_k) \neq \emptyset$. Let $s_{j_0} \in \mathrm{Comp}(\tilde{s}_{i_0}) \cap \mathrm{Comp}(\tilde{s}_k)$, i.e., $s_{j_0}$ is exogenously connected to both $x_{i_0}$ and $x_k$, where $x_{i_0}$ does not have unique components.

Note that (27)-(29) hold for this case as well; (27)-(29) do not depend on whether $x_{i_0}$ has unique components. When $x_{i_0}$ has no unique components, and one of its components $s_{j_0}$ is exogenously connected to $x_k$, this results in a similar indistinguishably problem as in the previous case. In particular, we are not able to distinguish between the part of $s_{j_0}$ component in $x_k$ that results from the causal connection/path from $x_{i_0}$, and the part that results from the exogenous connection from $s_{j_0}$. Thus, given $\tilde{\mathbf{W}}$, we can not distinguish the true model from the two models in (30), (31).

The only difference from case (i) is that $x_{i_0}$ does not have unique components. In this case, we cannot have any of the components in $\tilde{s}_{i_0}$ to be exogenously connected to $x_k$. In particular, all the shared (non-unique) components of $\tilde{s}_{i_0}$ are pivotal to recover the total causal effect from $x_{i_0}$ to $x_k$ and including any of them in $\tilde{s}_k$ will always result in indistinguishably of the generating model.

**Case (iii).** Suppose the marriage condition is not satisfied for the collection $\tilde{\mathcal{S}}_k$. Then as described in Appendix E.1, the matrix $\mathbf{B}_k$ is not of full row rank, where $\mathbf{B}_k$ is defined as $\mathbf{B}_k = [\mathbf{B}_{ij}]_{i,j: \, x_i \in \mathcal{P}_k, \, s_j \in \mathcal{E}_k}$.

Given the recovered mixing matrix $\tilde{\mathbf{W}}$, suppose that the exogenous connections to $x_k$, i.e., $\tilde{s}_k$, are recovered as in (26). Then the remaining source components in $x_k$ that result from the causal connections/paths from the possible parents in $\mathcal{P}_k$ can be written as

$$\sum_{s_j \in \mathrm{Comp}(x_k)} \tilde{w}_{kj} s_j - \tilde{s}_k = \sum_{s_j \in \mathcal{E}_k} \tilde{w}'_{kj} s_j = \sum_{i: x_i \in \mathcal{P}_k} \tilde{a}_{ki} \tilde{s}_i,$$

where $\{\tilde{w}_{kj}\}_{j: s_j \in \mathrm{Comp}(x_k)}$ are the entries corresponding to $x_k$ in $\tilde{\mathbf{W}}$ (i.e., $x_k = \sum_{s_j \in \mathrm{Comp}(x_k)} \tilde{w}_{kj} s_j$). $\{\tilde{w}'_{kj}\}$ are also known, as they result from subtracting the components of $\tilde{s}_k$ (known by assumption) from $x_k$. Our task is to recover the total causal effect $\tilde{a}_{ki}$ from each possible parent $x_i \in \mathcal{P}_k$ to $x_k$, which translates to finding the solution of

$$\sum_{i: x_i \in \mathcal{P}_k} \tilde{a}_{ki} \tilde{s}_i = \sum_{s_j \in \mathcal{E}_k} \tilde{w}'_{kj} s_j. \tag{32}$$

If we write each source mixture $\tilde{s}_i$ in terms of its source components, then (32) can be written as

$$\sum_{i:x_i \in \mathcal{P}_k} \tilde{a}_{ki} \sum_{s_j \in \mathcal{E}_k} b_{ij} s_j = \sum_{s_j \in \mathcal{E}_k} \tilde{w}'_{kj} s_j. \tag{33}$$

(33) must hold for each source component $s_j \in \mathcal{E}_k$, which means

$$\sum_{i:x_i \in \mathcal{P}_k} \tilde{a}_{ki} b_{ij} = \tilde{w}'_{kj}, \quad \forall s_j \in \mathcal{E}_k \qquad \implies \qquad \mathbf{B}_k^\top \tilde{\mathbf{a}}_k = \tilde{\mathbf{w}}'_k, \tag{34}$$

where $\tilde{\mathbf{a}}_k = [\tilde{a}_{ki}]_{i:x_i \in \mathcal{P}_k}$, and $\tilde{\mathbf{w}}'_k = [\tilde{w}'_{kj}]_{j:s_j \in \mathcal{E}_k}$. Recall, $\mathbf{B}_k^\top$ is not of full column rank, hence (34) has infinite number of non-zero solutions. Denote these solutions as $\{\tilde{\mathbf{a}}_k^{(l)} : l = 1, 2, \cdots\}$. For each $l$, we obtain a different generating model of $x_k$ as

$$x_k = \sum_{i:x_i \in \mathcal{P}_k} \tilde{a}_{ki}^{(l)} \tilde{s}_i + \tilde{s}_k. \tag{35}$$

We conclude that, given the mixing matrix, there exist infinite number of generating models that we cannot distinguish from one another, i.e., the true generating model cannot be uniquely identified.

## Appendix F. Other proofs

### F.1. Proof of equivalence between two representations of linear DS-P-SCM

In this subsection, we prove that linear DS-P-SCM can be equivalently written as a linear P-SCM or a linear DS-SCM. That is, the submodel of linear P-SCM under distinct source assumption is equivalent to the submodel of linear DS-SCM under linear latent confounding and jointly independent sources.

First, we show that a linear DS-SCM with linear latent confounding and jointly independent sources can be written as a linear P-SCM with every observed variable associated with a distinct source. Denote the vector of observed variables as $X$. A linear DS-SCM with linear latent confounding and jointly independent sources can be written as

$$X = \mathbf{A}_{ol} X_l + \mathbf{A} X + S_d = \mathbf{A} X + [\mathbf{A}_{ol} \quad \mathbf{I}] \begin{bmatrix} X_l \\ S_d \end{bmatrix}. \tag{36}$$

$X_l$ represents the vector of jointly independent latent confounders; $S_d$ represents the vector of jointly independent distinct sources; $\mathbf{A}$ represents the causal connection among observed variables; $\mathbf{A}_{ol}$ represents the linear latent confounding. Note that each column in $\mathbf{A}_{ol}$ contains at least two non-zero entries because it is a confounder. Under acyclicity assumption, $\mathbf{A}$ can be converted to a strictly lower triangular matrix following the causal order among observed variables. Hence, (36) is equivalent to linear P-SCM in (2), where the adjacency matrix is $\mathbf{A}$, the exogenous connection matrix is $\mathbf{B} = [\mathbf{A}_{ol} \quad \mathbf{I}]$, and $S = [X_l; S_d]$. Since $\mathbf{I}$ is a submatrix of $\mathbf{B}$, this means that each observed variable is associated with a distinct source, i.e., a source that is not shared with any other observed variables.

Next, we show by construction that a linear P-SCM with distinct sources can be written as a linear DS-SCM with linear latent confounding and jointly independent sources. Consider a linear P-SCM, $X = \mathbf{A} X + \mathbf{B} S$. A distinct source that is associated to only one observed variable corresponds to a

column in $\mathbf{B}$ with only one non-zero entry. Since each observed variable is associated with a distinct source, the number of these columns is exactly equal to the number of observed variables, and these columns can be permuted and scaled into an identity matrix. The remaining columns in $\mathbf{B}$ must have two or more non-zero entries (same structure as $\mathbf{A}_{ol}$), and hence each of these columns corresponds to a latent confounder where the confounding is linear. Thus, the linear P-SCM reduces to (36). This concludes the proof.

### F.2. Proof of equivalence between two representations of linear P-SCM

In this subsection, we prove that linear P-SCM (cf. (2)) can be equivalently written as (4), which is considered in most works studying linear DS-P-SCM. That is, instead of allowing an observed variable to have no distinct source, we can require the observed variable to have a distinct source but the source can have zero variance.

Notice that (4) can be equivalently written as

$$X = \mathbf{A}_{ol}X_l + \mathbf{A}X + \mathbf{B}_{oo}S_o, \tag{37}$$

where $S_o$ is the vector of distinct sources with non-zero variance. $\mathbf{B}_{oo}$ is composed of one-hot column vectors (i.e., binary vectors with exactly one non-zero entry) representing the correspondence between observed variables in $X$ and their associated non-zero distinct sources. If deterministic relation are not present (i.e., the distinct sources are all with non-zero variance), then $\mathbf{B}_{oo}$ is identity, and (4) is the same as (37). If some of the distinct sources have zero variance, then there are less sources in $S_o$ than observed variables. In this case, we can always rewrite (4) as (37) by removing the sources in $S_d$ with zero variance, and remove the corresponding column vectors in $\mathbf{I}$. We can also rewrite (37) as (4) by adding a distinct source with zero variance for each observed variable that is not associated with a non-zero distinct source in (37), and remove $\mathbf{B}_{oo}$.

To show that (37) reduces to a linear P-SCM, we have

$$X = \mathbf{A}X + \begin{bmatrix} \mathbf{A}_{ol} & \mathbf{B}_{oo} \end{bmatrix} \begin{bmatrix} X_l \\ S_o \end{bmatrix}. \tag{38}$$

Assuming the model is acyclic, $\mathbf{A}$ can be permuted into a strictly lower triangular matrix, hence (38) reduces to a linear P-SCM.

Next, we show that a linear P-SCM can be written as (37). Consider the linear P-SCM, $X = \mathbf{A}X + \mathbf{B}S$, where some observed variables are not associated with distinct sources. Recall from Appendix F.1 that a distinct source that is associated with only one observed variable corresponds to a column in $\mathbf{B}$ with one non-zero entry. However, the number of these columns is less than the number of observed variables, and hence these columns can not be permuted to identity. Thus, we can write $\mathbf{B}$ as $\begin{bmatrix} \mathbf{A}_{ol} & \mathbf{B}_{oo} \end{bmatrix}$, where each column of $\mathbf{B}_{oo}$ have exactly one non-zero entry, yet $\mathbf{B}_{oo}$ have more rows than columns, and hence some of its rows are zero. Columns of $\mathbf{A}_{ol}$ have two or more non-zero entries and still represent linear latent confounding. After we scale the non-zero entries in $\mathbf{B}_{oo}$ to 1 (by changing the scales of the sources in $S$ corresponding to these columns), the linear P-SCM reduces to (38), which can be further translated to (37).

### F.3. Proof of equivalence between two representations of the marriage condition

We first show that the marriage condition, when defined over the collection $\{\mathrm{Comp}(x_i) : x_i \in \mathcal{P}_k\}$, is equivalent to linear independence of the possible parents in $\mathcal{P}_k$ in the mixing matrix $\mathbf{W}$ (cf. (3)).

Recall that, for each $x_i \in \mathcal{P}_k$, $\mathrm{Comp}(x_i)$ includes the source components with indices corresponding to the non-zero entries in $x_i$'s row of $\mathbf{W}$.

Consider only the rows of $\mathbf{W}$ corresponding to possible parents in $\mathcal{P}_k$. The non-zero entries in these rows correspond to the existing source components, i.e., $\mathcal{E}_k = \bigcup_{x_i \in \mathcal{P}_k} \mathrm{Comp}(x_i)$ (cf. Definition 8). Let $X_k = [x_i]_{x_i \in \mathcal{P}_k}$ and $S_k = [s_j]_{s_j \in \mathcal{E}_k}$. Then we have

$$X_k = \mathbf{W}_k S_k, \quad \text{where } \mathbf{W}_k = [\mathbf{W}_{ij}]_{i,j: x_i \in \mathcal{P}_k, s_j \in \mathcal{E}_k}. \tag{39}$$

The following lemma shows the connection between the marriage condition and the rank of the submatrix $\mathbf{W}_k$.

**Lemma 11 (Edmonds (1967))** *Let $\mathbf{W}$ be an $m \times n$ matrix, where $m \leq n$, such that any submatrix (with non-zero rows or columns) of an arbitrarily permuted version of $\mathbf{W}$ is of full rank. Then, $\mathbf{W}$ has rank $m$ if and only if for every subset of $C$ rows, the corresponding submatrix of $\mathbf{W}$ has at least $C$ non-zero columns.*

According to Lemma 11, the marriage condition on the collection $\{\mathrm{Comp}(x_i) : x_i \in \mathcal{P}_k\}$ is equivalent to the matrix $\mathbf{W}_k$ in (39) being of full row rank. That is, by applying Lemma 11 to $\mathbf{W}_k$, for $\mathbf{W}_k$ to be of full rank, every subset of $C$ rows of $W_k$ (i.e., every $C$ possible parents) must have at least $C$ non-zero columns (i.e., $C$ different source components). $\mathbf{W}_k$ is full rank means that each possible parent is linearly independent from the other possible parents in $\mathcal{P}_k$.

Note that the recovered mixing matrix $\tilde{\mathbf{W}}$ from BSS is a column-repermuted and rescaled version of $\mathbf{W}$, which preserves the rank of the submatrices of $\mathbf{W}$. Thus the marriage condition is also equivalent to the linear independency among possible parents in the representation from BSS (as stated in Example 8).

Similarly, for each possible parent $x_i \in \mathcal{P}_k$, $\mathrm{Comp}(\tilde{s}_i)$ includes the source components with indices corresponding to the non-zero entries in $x_i$'s row of $\mathbf{B}$. Since each $x_i \in \mathcal{P}_k$ can be written as a linear combination of the source mixtures $\tilde{s}$ in its parent set (plus $\tilde{s}_i$), all of which are included in $\tilde{\mathcal{S}}_k = \{\tilde{s}_i : x_i \in \mathcal{P}_k\}$, we have $\mathcal{E}_k = \bigcup_{x_i \in \mathcal{P}_k} \mathrm{Comp}(x_i) = \bigcup_{\tilde{s}_i \in \tilde{\mathcal{S}}_k} \mathrm{Comp}(\tilde{s}_i)$. Therefore, similar to (39), by considering only the rows of $\mathbf{B}$ that correspond to the possible parents in $\mathcal{P}_k$, the non-zero entries in these rows correspond to $\mathcal{E}_k$.

$$\mathbf{B}_k = [\mathbf{B}_{ij}]_{i,j: x_i \in \mathcal{P}_k, s_j \in \mathcal{E}_k}. \tag{40}$$

Once again, applying Lemma 11 to matrix $\mathbf{B}_k$, the marriage condition on the collection $\{\mathrm{Comp}(\tilde{s}_i) : \tilde{s}_i \in \tilde{\mathcal{S}}_k\}$ is equivalent to the full row rank of $\mathbf{B}_k$.

Note that the parents of $x_i \in \mathcal{P}_k$ are also possible parents of $x_k$ (belong to $\mathcal{P}_k$). Thus, the rows of $\mathbf{A}$ that correspond to possible parents of $x_k$ have non-zero entries only in the columns of $\mathbf{A}$ that also correspond to the possible parents of $x_k$. Therefore, $X_k$ can be written as

$$X_k = \mathbf{A}_k X_k + \mathbf{B}_k S_k, \tag{41}$$

where $\mathbf{A}_k = [\mathbf{A}_{ij}]_{i,j: x_i \in \mathcal{P}_k, x_j \in \mathcal{P}_k}$ is the submatrix of $\mathbf{A}$ with both rows and columns corresponding to possible parents of $x_k$. Comparing (39) and (41), we have

$$\mathbf{W}_k = (\mathbf{I} - \mathbf{A}_k)^{-1} \mathbf{B}_k. \tag{42}$$

Since $\mathbf{A}_k$ can be permuted to a strictly lower triangular matrix, $\mathbf{I} - \mathbf{A}_k$ is of full rank. Thus $\mathbf{W}_k$ is of full row rank if and only if $\mathbf{B}_k$ is of full row rank. To conclude, the marriage condition defined over the collection of the component sets of possible parents, is equivalent to the condition over the collection of the component sets of their source mixtures.

### F.4. Proof of Theorem 6

We have mentioned in Section 4.1 that if each observed variable in the linear P-SCM is associated with a distinct source, then for each observed variable $x_k$, the possible parent set is exactly the ancestor set of $x_k$, and every possible parent $x_i \in \mathcal{P}_k$ has a unique component (i.e., $U_k(i) \neq \emptyset$). Besides, for each observed variable $x_i$, the component set of its mixtures, i.e. $\mathrm{Comp}(\tilde{s}_i)$, includes a distinct source. Thus, for any subset $X_C$ of the observed variables in $\mathcal{X}$, $\left| \bigcup_{i:x_i \in X_C} \mathrm{Comp}(\tilde{s}_i) \right| \geq |X_C|$. This implies that the marriage condition (Condition 2) holds for every observed variable $x_k$, where we select the collection $X_C$ as any subset of the possible parent set $\mathcal{P}_k$.

Therefore, a reduced linear P-SCM could be uniquely identified from the recovered mixing matrix $\tilde{\mathbf{W}}$ if and only if for each observed variable $x_k$, the unique components condition (Condition 1) is satisfied: For every exogenous connection from $s_j$ to $x_k$, it must belong to one of the two following cases:

- $s_j$ is a new component that is not exogenously connected to any of the possible parents of $x_k$. This means that no observed variables containing $s_j$ are possible parents of $x_k$, thus there are no causal paths from any observed variable containing $s_j$ to $x_k$.

- $s_j$ is connected to some possible parents in $\mathcal{P}_k$. Condition 1 implies that $s_j$ cannot be a unique component of any $x_i \in \mathcal{P}_k$, and it must be exogenously connected to at least two observed variables $x_{i_1}, x_{i_2}$ in $\mathcal{P}_k$, where $i_1 \neq i_2$. Since $\mathcal{P}_k$ is the ancestor set of $x_k$, there are at least two distinct causal paths $x_{i_1} \rightsquigarrow x_k$ and $x_{i_2} \rightsquigarrow x_k$ from the observed variables with $s_j$ in their source mixtures (i.e., exogenously connected to $s_j$) to $x_k$.

**Remark 12** *In a graphical representation of the reduced linear P-SCM, the condition in Theorem 6 can be expressed as follows. Let us define a causal cycle as an undirected cycle in the graph, where one directed edge on this cycle is from a source variable $s_j$ to an observed variable $x_k$, and the remaining edges of the cycle form a directed path from $s_j$ to $x_k$. We call the source $s_j$ and the observed variable $x_k$ as the source and sink of the causal cycle. Then the reduced linear P-SCM is uniquely identifiable if and only if for each causal cycle in the graph, there exists at least one other causal cycle that shares the same source and sink (including the exogenous connection between them).*

*A simpler representation of the condition (which is not generally necessary and sufficient) is that: The generating model is not uniquely identifiable if the graphical representation includes a causal cycle, and none of the edges on this cycle is shared with other cycles in the graph.*

### F.5. Proof of Remark 7

For the sake of completeness, we state the following theorem.

**Lemma 13 (Theorem 16, Salehkaleybar et al. (2020))** *Let $des_o(x_i)$ be the observed descendant set of variable $x_i$ in a linear DS-P-SCM with latent confounders (including $x_i$ itself if it is observed).*

*Then the linear DS-P-SCM is uniquely identifiable from observations (i.e., the total causal effect between any two observed variables can be identified), if for any observed variable $x_i$ and any latent variable $x_k$, $des_o(x_i) \neq des_o(x_k)$.*

As we explained in Appendix F.1, a latent confounder in the linear DS-P-SCM corresponds to a shared source that is connected to at least two observed variables in the representation of linear P-SCM (with distinct source). In the following, we first show that the condition in Theorem 6 implies the condition in Lemma 13. Note that the observed descendant set of a latent confounder and that of an observed variable must be different if both variables are not causally connected. Therefore, it suffices to show that if the condition in Theorem 6 is satisfied, then for a source $s_j$ that is exogenously connected to at least two observed variables including $x_k$, $des_o(s_j) \neq des_o(x_k)$.

According to Theorem 6, since $s_j$ is exogenously connected to $x_k$, one of the following situations must hold:

- There are at least two distinct causal paths from observed variables (which are exogenously connected to $s_j$) to $x_k$. Suppose $x_i$ is an ancestor of $x_k$ and is exogenously connected to $s_j$. Then we have $x_i \in des_o(s_j)$, and $x_i \notin des_o(x_k)$. Thus $des_o(s_j) \neq des_o(x_k)$.

- There are no causal paths from observed variables containing $s_j$ to $x_k$. Recall that $s_j$ is exogenously connected to at least one observed variable other than $x_k$. Denote $x_l$ as the observed variable exogenously connected to $s_j$ that has the smallest index in the causal order among observed variables (other than $x_k$). That is, all observed variables that are exogenously connected to $s_j$ must follow $x_l$ in the causal order except for $x_k$. In this case we have $x_l \in des_o(s_j)$. In the following we show that $x_l \notin des_o(x_k)$, which means that $des_o(s_j) \neq des_o(x_k)$, and completes the proof.

  The proof follows by contradiction. Suppose $x_l \in des_o(x_k)$. This means that there is a causal path from $x_k$ to $x_l$. Now consider $x_l$. Since $s_j$ is connected to both $x_k$ and $x_l$, according to Theorem 6, there must exist another causal path from an observed variable (exogenously connected to $s_j$) to $x_l$. However, such an observed variable must precede $x_l$ in the causal order, which contradicts the fact that $x_l$ has the smallest index among all observed variables connected to $s_j$ other than $x_k$. Therefore $x_l \notin des_o(x_k)$.

To show the second part of the remark, i.e., the condition in Lemma 13 does not imply the condition in Theorem 6, we provide an example which satisfies the condition in Lemma 13 but does not satisfy the condition in Theorem 6. Hence, the generating model in this example is uniquely identifiable among all linear DS-P-SCMs, but is not uniquely identifiable among all linear P-SCMs.

**Example 9** *Consider the following linear DS-P-SCM with three observed variables $x_1$, $x_2$, $x_3$ and a latent confounder $x_L$:*

$$x_L = s_L; \quad x_1 = a_{1L}x_L + s_1;$$
$$x_2 = a_{2L}x_L + s_2; \quad x_3 = a_{3L}x_L + a_{31}x_1 + s_3. \tag{43}$$

*The graphical representation of this DS-P-SCM is shown in the plot on the left in Figure 8. The observed descendant sets of all four variables are*

$$des_o(x_L) = \{x_1, x_2, x_3\}; \quad des_o(x_1) = \{x_1, x_3\};$$
$$des_o(x_2) = \{x_2\}; \quad des_o(x_3) = \{x_3\}.$$

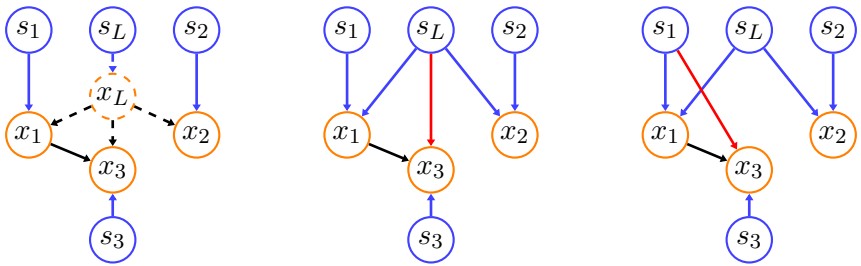

Figure 8: (Left) The ground truth linear DS-P-SCM for Example 9. This DS-P-SCM satisfies the condition in Lemma 13. (Middle) The ground truth can be equivalently modeled as a linear P-SCM. This P-SCM does not satisfy the condition in Theorem 6. (Right) An observationally equivalent linear P-SCM, which can not be modeled as a linear DS-P-SCM. Red arrows represent the differences.

*Therefore, all four variables have distinct observed descendant sets, which satisfies the condition in Lemma 13. Hence this linear DS-P-SCM is uniquely identifiable.*

*The linear DS-P-SCM in* (43) *can be equivalently written into the following linear P-SCM:*

$$x_1 = \tilde{s}_1 = a_{1L}s_L + s_1; \quad x_2 = \tilde{s}_2 = a_{2L}s_L + s_2;$$
$$x_3 = a_{31}x_1 + \tilde{s}_3, \ \tilde{s}_3 = a_{3L}s_L + s_3. \tag{44}$$

*The graphical representation of this linear P-SCM is shown in the plot in the middle in Figure 8. This model violates the condition in Theorem 6: $s_L$ is exogenously connected to $x_3$, but there is only one other path from an observed variable containing $s_L$ to $x_3$ (i.e., $x_1 \rightarrow x_3$). Thus, this linear P-SCM is not uniquely identifiable, and an alternative linear P-SCM is shown in the plot on the right in Figure 8. The generating models of $x_1$ and $x_2$ in this alternative P-SCM are the same as in* (44), *but the generating model of $x_3$ is*

$$x_3 = (a_{31} + a_{3L}/a_{1L})x_1 + \tilde{s}_3; \quad \tilde{s}_3 = -(a_{3L}/a_{1L})s_1 + s_3.$$

*Notice that this linear P-SCM cannot be represented as a linear DS-P-SCM, since there is no distinct source associated with $x_1$. Hence it does not affect the unique identifiability of the generating model in the class of linear DS-P-SCMs.*

## Appendix G. Numerical experiments details

We generate synthetic models according to a linear P-SCM with $p$ observed variables and $rp$ source variables, where $r$ is the ratio of source to observed variables.[9] Each pair of observed variables is connected with probability $d_e/(p-1)$, and each pair of source and observed variable is connected with probability $d_o/p$. $d_e$ (resp. $d_o$) is the average number of causal (resp. exogenous) connections for an observed (resp. a source) variable. That is, on average, each observed variable is causally connected to $d_e$ other observed variables, and each source variable is exogenously connected to $d_o$ observed variables. We refer to $r$, $d_e$, and $d_o$ as the source to observed node ratio, average observed node degree, and average source node degree, respectively. The strength of each connection is drawn

---

9. We use the words "variable" and "node" interchangeably, to refer to an observed/source variable.

---

**Algorithm 2:** Verifying satisfiability of Conditions 1 and 2 for a linear P-SCM

---

**Input:** Adjacency matrix $\mathbf{A}$, and exogenous connection matrix $\mathbf{B}$ of a linear P-SCM

1  Compute $\mathbf{W} = (\mathbf{I} - \mathbf{A})^{-1}\mathbf{B}$;
2  Repermute $\mathbf{W}$ such that the number of non-zero entries in each row is in an increasing order
   (rows with equal number of non-zero entries are permuted at random) ;
3  Repermute $\mathbf{A}$ and $\mathbf{B}$ according to the order derived in Step 2;
4  **for** $k = 1 : p$ **do**
5     Find the possible parent set $\mathcal{P}_k$ using $\mathbf{W}$;
6     Using $\mathbf{A}[k, :]$, verify that all parents of $x_k$ are included in $\mathcal{P}_k$, otherwise abort the algorithm
      since the model does not satisfy Assumption 2;
7     Initialize $\bar{\mathcal{P}}_k = \mathcal{P}_k$;
8     Find the set $\mathcal{U}$ of possible parents in $\bar{\mathcal{P}}_k$ that have unique components ;
9     **if** $|\mathcal{U}| \neq 0$ **then**
10       Using $\mathbf{B}[k, :]$; verify that $x_k$ is not connected to any unique components of $x_i$ for each
         $x_i \in \mathcal{U}$, otherwise abort the algorithm ;
11       Update $\bar{\mathcal{P}}_k$ as $\bar{\mathcal{P}}_k \leftarrow \bar{\mathcal{P}}_k \setminus \mathcal{U}$; Go back to Step 8, using the updated $\bar{\mathcal{P}}_k$ ;
12    **else**
13       Set $\mathcal{I}_k = \bar{\mathcal{P}}_k$, i.e., $\mathcal{I}_k$ is the set of possible parents with no unique components after any
         number of iterations;
14       Select the rows of $\mathbf{B}$ that correspond to the possible parents in $\mathcal{I}_k$, denote these rows by
         $\mathbf{B}_I$ ;
15       Verify that the number of non-zero columns in $\mathbf{B}_I$ is greater than or equal the number of
         its rows, otherwise abort the algorithm ;
16       Using $\mathbf{B}[k, :]$, verify that $x_k$ is not connected to any sources corresponding to the
         non-zero columns of $\mathbf{B}_I$, otherwise abort the algorithm ;

**Output:** If verification steps 10, 15, 16 yield true for all $k$, the model satisfies our conditions.

---

independently from a uniform distribution over the interval $[-1, -0.5] \cup [0.5, 1]$. We randomly permute both the observed and source variables to hide the true causal order after model generation.

### G.1. Satisfiability of the conditions

To test the restrictiveness of Conditions 1 and 2, i.e., the unique component and marriage conditions in Sections 3.1 and 3.2, we propose Algorithm 2 which verifies whether the generating model satisfies these conditions. Given matrices $\mathbf{A}$ and $\mathbf{B}$ of a linear P-SCM, Algorithm 2 first computes the true mixing matrix $\mathbf{W}$, and use it to retrieve the possible parent set for each observed variable $x_k$. Subsequently, the algorithm implements the same iterative procedure as in Definition 3 to verify that the unique components condition is satisfied.

From Appendix E.1, the marriage condition can be equivalently applied to the set of possible parents with no unique components, i.e., $\mathcal{I}_k$. The marriage condition then translates to: For any $X_C \subseteq \mathcal{I}_k$, $|X_C| \leq |\bigcup_{i:x_i \in X_C} \mathrm{Comp}(\tilde{s}_i)|$. Note that, Algorithm 2 only verifies the satisfiability of this condition for the subset $X_C = \mathcal{I}_k$. Thus, the condition verified in Algorithm 2 is weaker than the marriage condition; yet it is much more efficient to verify. Indeed, we do not observe much difference in the results when we only verify the marriage condition for $X_C = \mathcal{I}_k$, as opposed to verifying the condition for all $X_C \subseteq \mathcal{I}_k$.

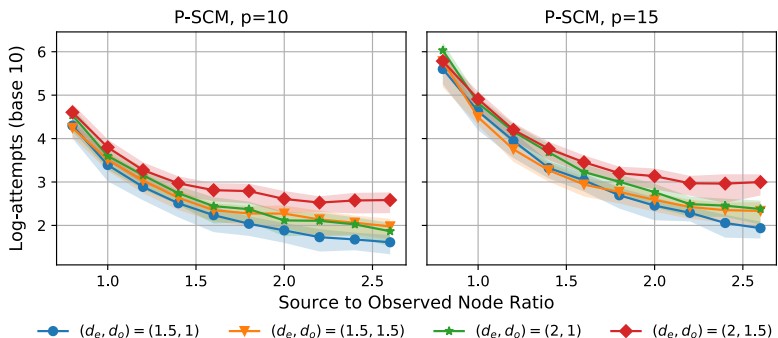

Figure 9: Satisfiability of our conditions for models generated according to a linear P-SCM. $d_e$ is the average observed node degree, and $d_o$ is the average source node degree.

**Satisfiability of P-SCMs.** We first test the satisfiability of the conditions with different source to observed node ratios and different average degrees. We select $p = 10, 15$ observed variables, average observed node degree $d_e = 1.5, 2$, and average source node degree $d_o = 1, 1.5$. Under each setting, we set the source to observed node ratio $r$ to be ranging from 0.8 to 2.6, with a step size of 0.2. We report the number of generated models (attempts) required to obtain a model that satisfies our conditions. We repeat the experiment 100 times, and calculate the average number of attempts (to find a satisfying model) and the standard deviations. The common logarithms (base 10) of average # attempts are shown in Figure 9. The shaded area around each curve represents the logarithm of average # attempts plus/minus half the standard deviation.

We observe that when there are less or equal number of sources than observed variables, our conditions are unlikely to be satisfied. This is due to the high probability of overlap among the exogenous connections, i.e., the same source is connected to multiple observed variables. This leads to a high probability of the model violating the unique component condition. When the ratio $r$ increases, the sources are less likely to be connected to multiple observed variables, hence the probability of satisfiability increases, converging to a fixed value.

Now, let us compare the # attempts for different average degrees within each subplot ($p = 10$ and $p = 15$). We observe when $r$ is large (greater than 2), increasing the observed node degree, $d_e$, is more likely to result in complex structures (among observed nodes). Further, increasing the source node degree, $d_o$, implies that more sources are shared among observed variables. Thus, increasing $d_e$ and $d_o$ increases the # attempts required to obtain a satisfying model. When $r$ is small (less than 1), increasing $d_e$ again results in a complex structure. However, since the number of source variables is small, a small $d_o$ implies that observed variables are more likely to have either no exogenous connections, or a few that are shared among them. Subsequently, when $r$ is small, the average # attempts required to generate a satisfying model is large for large $d_e$ and small $d_o$. This is shown by the green line in both subplots. We conclude that the satisfiability of our conditions depends on:

1. The existence and similarity of exogenous connections among observed variables,

2. The ratio of overlap among exogenous connections (i.e., how many sources are shared among observed variables),

3. The structure of the causal graph among observed variables.

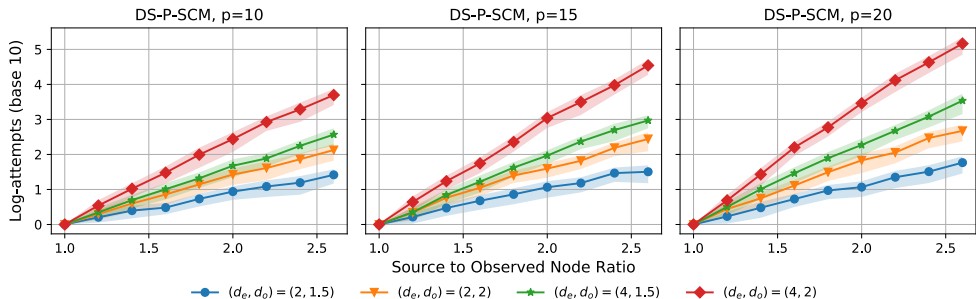

Figure 10: Satisfiability of our conditions for models generated according to a linear DS-P-SCM.

**Satisfiability of DS-P-SCMs.** To further demonstrate our ideas, we test the satisfiability of the conditions when each observed variable in the generating model has a distinct source that is not shared with any other observed variable. As discussed in Section 4, in this case, the model is a linear DS-P-SCM. We select $p = 10, 15, 20$ observed nodes, $d_e = 2, 4$, and $d_o = 1.5, 2$. We report the average and the standard deviation of the # attempts to generate 100 satisfying models. Figure 10 shows the logarithm of the # attempts for the ratio $r$ ranging from 1 to 2.6. When $r = 1$, the model can be exactly represented as a linear DS-P-SCM without latent confounding. Indeed, all models generated under this setting satisfy our conditions (satisfiability probability is equal to 1). Further, we observe that the # attempts needed to obtain satisfying models increases as $r$ increases. Also, it is more likely for sparser graphs (small $d_e$ and $d_o$) to satisfy our conditions than denser graphs (large $d_e$ and/or $d_o$). These observations are aligned with our aforementioned conclusions.

### G.2. Performance of recovery

**Uniquely identifiable cases.** We test the performance of our P-SCM recovery algorithm (Algorithm 1) for models generated according to each of the following settings:

(1) P-SCM with equal number of observed variables and sources, and $(d_e, d_o) = (1.5, 1.5)$. We refer to this setting as `P-SCM_Equal`.

(2) P-SCM with 2 less sources than observed variables, and $(d_e, d_o) = (1.5, 1.5)$. We refer to this setting as `P-SCM_Fewer`.

(3) DS-P-SCM with 3 more sources than observed variables, and $(d_e, d_o) = (2, 1.5)$. We refer to this setting as `DS-P-SCM`.

We select only the generated models that satisfy our conditions, and hence are uniquely identifiable. The sources are independent and identically distributed according to a uniform distribution over the interval $[-0.5, 0.5]$. We generate $n = 1000$ samples of observations. For `P-SCM_Equal` and `P-SCM_Fewer`, we use FastICA (Hyvarinen, 1999) for blind source separation, and for `SCM`, we use ReconstructionICA (Le et al., 2011).

We prune the mixing matrix using bootstrapping method (Efron and Tibshirani, 1994). We first generate $n_{boot} = 50$ bootstrap samples by sampling from the observed data. Then, for each bootstrap sample, we use ICA method to deduce estimates $\tilde{\mathbf{W}}^{(i)}$, $i = 1, \cdots, n_{boot}$ of the mixing matrix. Recall that each estimate suffers from permutation and scaling indeterminacies. For each estimate $\tilde{\mathbf{W}}^{(i)}$, we normalize each column by the entry with the largest absolute value on that column. Further, we

permute the columns of each $\tilde{\mathbf{W}}^{(i)}$, $i = 2, 3, \cdots, n_{boot}$ such that the Frobenius distance between the permuted $\tilde{\mathbf{W}}^{(i)}$ and $\tilde{\mathbf{W}}^{(1)}$ is minimized. Lastly, we compute the average of the bootstrap estimates (samples) of the mixing matrix, and prune this average by applying a Student t-test element-wise in order to reject the entries with small variance around zero. In particular, we keep each entry with 95% confidence level. The recovered mixing matrix $\tilde{\mathbf{W}}$ is the resulting matrix after pruning the average of the bootstrap estimates. Notice that we assume that the number of sources is known for ICA recovery.

We evaluate the performance of ICA recovery by comparing the recovered mixing matrix $\tilde{\mathbf{W}}$ and the actual mixing matrix $\mathbf{W}$, and call the ICA recovery successful when $\tilde{\mathbf{W}}$ has the same non-zero entries (support) as $\mathbf{W}$ (after column permutations). That is, ICA learns the exact structure of $\mathbf{W}$ from observed data.

We evaluate the performance of our P-SCM Recovery algorithm using the recovered mixing matrix $\tilde{\mathbf{W}}$ from ICA, and compare the performance with four other algorithms: ICA-LiNGAM[10] (Shimizu et al., 2006), DirectLiNGAM (Shimizu et al., 2011), Pairwise lvLiNGAM (Entner and Hoyer, 2010) and ParceLiNGAM (Tashiro et al., 2014), whenever applicable. These baseline methods (i) are all based on LiNGAM (or lvLiNGAM) and (ii) can recover the strengths of causal connections between observed variables, i.e., matrix $\mathbf{A}$. We implement all baseline methods using the codes released by their corresponding authors, and select default hyperparameters throughout the simulation.

A well-known class of methods for causal structure learning in the presence of latent confounders is FCI algorithm (Spirtes et al., 2000) and its variants. However, we do not compare our approach with these methods since these methods returns a PAG, which is a class of Maximal Ancestral Graphs (MAGs), and our approach mostly focuses on unique identification of the ground truth (i.e., DAG). First of all, since our hypothesis space of possible graphical models (linear P-SCMs) is different from the class considered by FCI (DS-SCMs), the returned PAG may not contain the ground truth even if FCI does not make any mistakes in the estimation, and hence we do not believe that comparison is fair. Secondly, we note that in a MAG, two variables can be adjacent even if in the ground truth DAG they are not. Therefore, it is not straightforward to define a measure of performance between a MAG and the ground truth DAG. Third, we note that the size of the PAG returned by FCI can be exponentially large and hence, even if we have an evaluation method for each MAG, still computation time can make the comparison infeasible.

We replace the step of solving an overdetermined linear system (Step 10 in Algorithm 1) by finding its least-squares approximation, in order to improve stability. We prune the recovered matrices $\mathbf{A}$, by removing the edges with causal strength $|a_{ij}| < 0.1$ for all methods including ours. We compare the differences between the recovered adjacency matrix, and the true adjacency matrix $\mathbf{A}$, using the following four test metrics:

(1) SHD / Edge; the normalized structural Hamming distance divided by the number of edges in the true model;

(2) Frobenius Norm;

(3) "Precision": The fraction of edges in the recovered model that appear in the true model;

(4) "Recall" (true positive rate): The fraction of edges in the true model that are correctly recovered.

---

10. In this subsection, we refer to the LiNGAM algorithm proposed by (Shimizu et al., 2006) as ICA-LiNGAM, in order to distinguish this method from others. Note that the other three baseline methods do not use ICA.

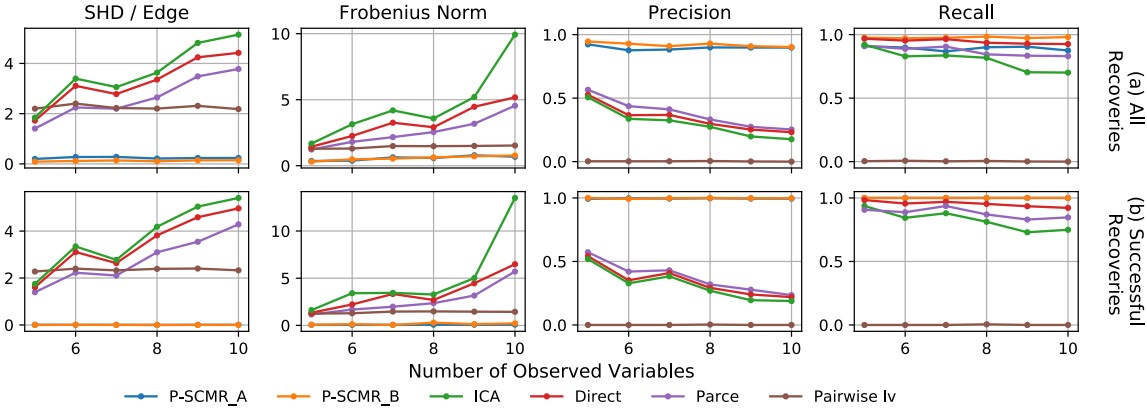

Figure 11: Performance of algorithms under `P-SCM_Equal` setting. For the successful recoveries case, the P-SCMR_A (blue line) overlaps with the P-SCMR_B (orange line) in all four metrics.

Note that for SHD / Edge and Frobenius Norm, lower values mean better performance. For Precision and Recall, higher values mean better performance.

We also evaluate the performance of our P-SCM Recovery algorithm for recovering the matrix $\mathbf{B}$, up to permutation and scaling indeterminacies, i.e., $\tilde{\mathbf{B}}$. We normalize both $\mathbf{B}$ and $\tilde{\mathbf{B}}$ such that the entry with the maximum absolute value on each column is one. We then permute the columns of $\tilde{\mathbf{B}}$ by minimizing the Frobenius distance:

$$\tilde{\mathbf{B}}^* = \arg\min_{\mathbf{C} = \tilde{\mathbf{B}}\mathbf{P}} ||\mathbf{B} - \mathbf{C}||_F, \quad \text{where } \mathbf{P} \text{ is a permutation matrix.}$$

We compare the difference between $\tilde{\mathbf{B}}^*$ and $\mathbf{B}$ using the same four metrics, and compare these results with the recovery of $\mathbf{A}$. We use P-SCMR_A and P-SCMR_B to represent the recovery of matrices $\mathbf{A}$ and $\mathbf{B}$ by our algorithm.

We repeat the simulation until 50 models can be successfully recovered by ICA, and report the average of the aforementioned four metrics. Figures 11 - 13 show the performance of recoveries for all algorithms under Settings (1)-(3) for model generation, with the number of observed variables ranging from 5 to 10. We compare the performance for (i) all ICA recoveries, and (ii) only the successful recoveries by ICA. We observe that for `P-SCM_Equal` and `P-SCM_Fewer`, our algorithm can learn the model more accurately than the existing algorithms. This is because existing algorithms may misinterpret the shared sources among observed variables as confounding or direct causal connections. We observe from the plots for SHD/Edge and Precision that this misinterpretation may add additional edges to the recovered graph, which do not exist in the true model. For the third setting, `DS-P-SCM`, the performance of our method is comparable to existing algorithms.

Besides, we observe that our algorithm performs significantly better than other algorithms when ICA recovery is successful, both in learning the structure of matrices $\mathbf{A}$, $\mathbf{B}$, and in learning the causal strengths (the coefficients). As discussed earlier, our algorithm assumes that the true mixing matrix $\mathbf{W}$ can be correctly recovered up to permutation and scaling of its columns. Thus, our algorithm depends on the accuracy of BSS recovery.

**Non-uniquely identifiable cases.** We evaluate the performance of our algorithm when the generating model does not satisfy the necessary and sufficient conditions for unique identifiability,

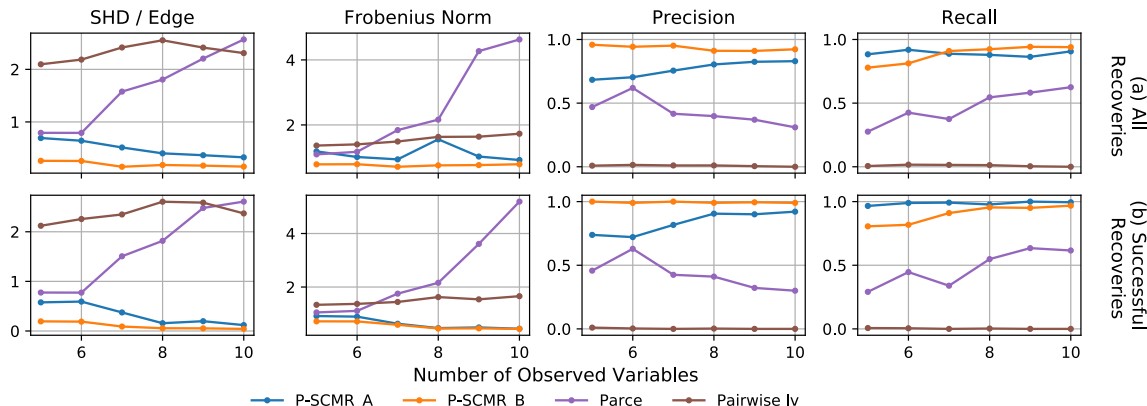

Figure 12: Performance of algorithms under `P-SCM_Fewer` setting. ICA-LiNGAM and DirectLiNGAM cannot be applied to this case, since we have fewer sources than observed variables.

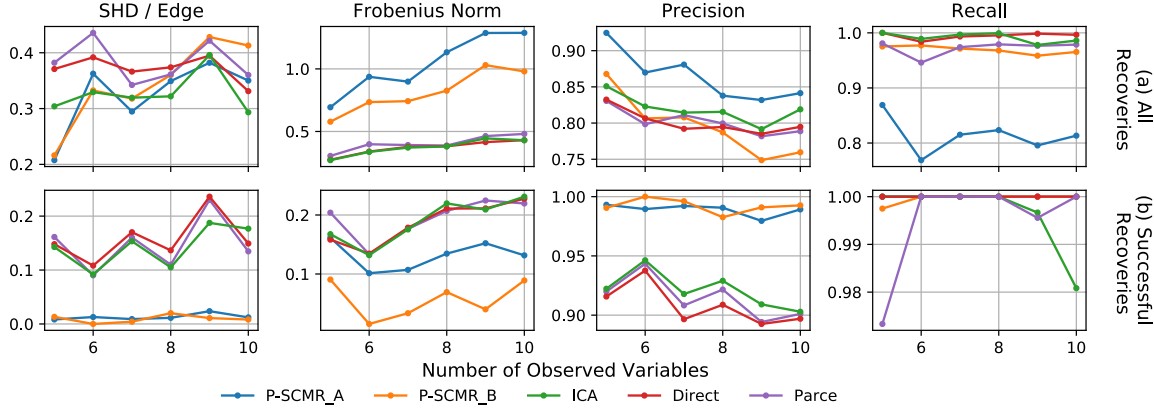

Figure 13: Performance of algorithms under `DS-P-SCM` setting. We ignore Pairwise lvLiNGAM since its performance is significantly worse than other methods, where it outputs empty graphs.

i.e., Conditions 1 and 2. Since the generating model cannot be uniquely recovered, there exist multiple generating models that correspond to the same mixing matrix $\mathbf{W}$. However, since our conditions are imposed separately on every observed variable $x_k$, our algorithm can recover part of the generating model where the conditions are satisfied. Here, we consider `P-SCM_Equal` setting (the number of observed variables is equal to the number of sources). We only select the generating models that do not satisfy our conditions, and refer to the modified setting as `P-SCM_NonUniq`. Figure 14 shows the recovery performance for our algorithm, compared with the baseline algorithms. Not surprisingly, for `P-SCM_NonUniq`, our algorithm does not perfectly recover the generating model, even when ICA recovery is successful (cf. Figure 14). However, our method outperforms the other algorithms in learning the structure and the causal strengths.

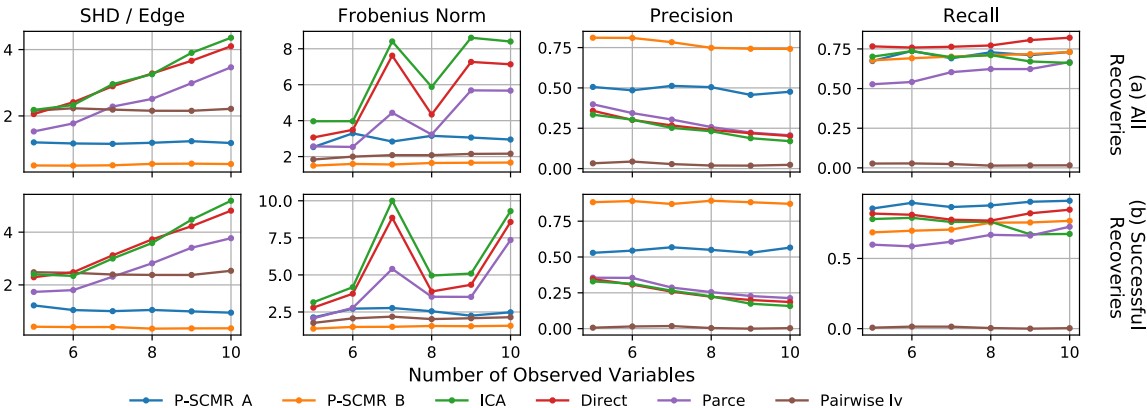

Figure 14: Performance of algorithms under `P-SCM_NonUniq` setting.

## G.3. Performance on real data

We test the performance of our recovery algorithm on the daily closing prices of the following five world stock indices from 01/01/2015 to 06/30/2020: (1) Dow Jones Industrial Average (DJI) in USA, (2) Nikkei 225 (N225) in Japan, (3) Euronext 100 (N100) in Europe, (4) Hang Seng Index (HSI) in Hong Kong, and (5) the Shanghai Stock Exchange Composite Index (SSEC) in China. We only select the days where all five stock markets are active; we have $T = 1192$ days for our observations. This data is obtained from Yahoo finance database[11]. The same setting, but with different data (different dates) has been considered in (Salehkaleybar et al., 2020, Section 5.2).

Let $c_i(t)$ be the closing price of the $i$-th index on day $t$. We define the corresponding return of an index $i \in \{\text{DJI, N225, N100, HSI, SSEC}\}$, at day $t$ as $R_i(t) = (c_i(t) - c_i(t-1))/c_i(t-1)$, for all $t = 2, 3, \cdots, T$. In this setting, we consider the corresponding return of an index $i$ to be an observed variable. We suppose these observed variables were generated according to a linear P-SCM with five sources, where the sources are non-Gaussian random variables.

We apply FastICA with bootstrapping for source separation from observations. The recovered mixing matrix $\tilde{\mathbf{W}}$ is

$$
\begin{bmatrix}
0.9096 & 0.2761 & 0 & 0 & 0 \\
0 & 0.7993 & 0 & 0.7414 & 0.2048 \\
0.4412 & 0.7738 & 0.1805 & -0.2962 & 0 \\
0.1537 & 0.4141 & 0.2902 & 0.1992 & 0.9398 \\
0.1480 & 0.2048 & 1.0000 & 0.4624 & 0.3513
\end{bmatrix},
\tag{45}
$$

where the five rows correspond to DJI, N225, N100, HSI and SSEC respectively. Using P-SCM Recovery algorithm, the directed graph among these observed variables can be recovered as in Figure 15.

For a directed acyclic graph (DAG), let the DAG-source node be the node with no parents, and the DAG-sink node be the node with no children. We observe from both $\tilde{\mathbf{W}}$ and Figure 15 that DJI is a DAG-source, with the fewest number of source components among all other indices. Further, we observe that DJI, N225 and N100 all have causal effects on HSI. Both observations are known to be true from common belief in economy, and from previous results in (Hyvärinen et al., 2010;

---

11. https://finance.yahoo.com

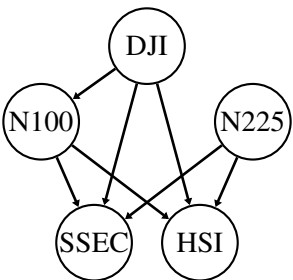

Figure 15: Recovered causal relations among five world stock indices using P-SCM Recovery algorithm.

Salehkaleybar et al., 2020). One main difference in our result from the result in (Hyvärinen et al., 2010) is that our algorithm cannot detect the directed edge from SSEC to HSI. This is because $\tilde{\mathbf{W}}$ in (45) has the same components for HSI and SSEC, and our algorithm concludes that there are no causal relation between these two observed variables. This may happen when the exogenous connections for these two variables are fully overlapped, as in this example.

Recall that our linear P-SCM is advantageous when there are more sources than observed variables. However, in this example, we are assuming a number of sources equal to the number of observed variables. This is mainly so that we are able to use FastICA, which provides a highly accurate estimate for the mixing matrix (when a ground truth is available). The existing overcomplete ICA methods on the other hand are not as accurate.

