# OpenReview forum: "Causal Discovery in Linear Structural Causal Models with Deterministic Relations"
_cclear.cc/CLeaR/2022/Conference — CLeaR 2022 Poster_

### Official Review · Reviewer_fNiS · 2021-11-18

**Confidence:** 4
**Overall Score:** 7

**Main Review:**

This paper considers the causal discovery problem in linear structural causal models, where the observed variables are of a deterministic function of other observed variables or latent confounders, but not with an exogenous source. The authors give necessary and sufficient conditions for the unique identifiability of the model and the relative estimation method. Overall, this paper is written well, with complete theoretical identifiability analysis for the model and sufficient supplementary materials. However, I have some confuses, listed below.

1. About the motivation of the introduced deterministic linear model, I agree that “G-SCMs without distinct sources can be used to model scenarios with less factors than observed nodes;”, but why deterministic relations are implied? For example, I mean in Fig.1(a), s_1 and s_2 can be removed while x_3 could have a distinct exogenous variable. I do not think the motivation is applicable for the proposed model. Btw, if it is indeed deterministic relation, the model in Eq. (2) cannot characterize this determinism.

2. I am confused about Assumption 2, which I think is contradictory with the proposed “deterministic relations”. For instance, if we consider a two-variable case and suppose x_i -> x_j, then according to Assumption 2, x_j has at least one source in x_j that is not in x_i, which implies x_j does not deterministically depend on x_i or latent confounders. Please explain it.

3. For the nonlinear causal discovery methods with deterministic relations and with identifiability, please also refer to the following papers.
[1] Chen Z, Zhang K, and Chan L, "Nonlinear Causal Discovery for High Dimensional Data: A Kernelized Trace Method," 2013 IEEE 13th International Conference on Data Mining, 2013, pp. 1003-1008, doi: 10.1109/ICDM.2013.103
[2] Zeng Y, Hao Z, Cai R, et al. Nonlinear Causal Discovery for High-Dimensional Deterministic Data[J]. IEEE Transactions on Neural Networks and Learning Systems, 2021.
[3] Janzing D, Mooij J, Zhang K, et al. Information-geometric approach to inferring causal directions[J]. Artificial Intelligence, 2012, 182: 1-31.

4. The sentence in the paper “when the sources are non-Gaussian random variables, the mixing matrix W can be recovered using Independent Component Analysis (ICA) or overcomplete ICA methods” is not correct, but using the under-complete ICA.

5. It might be better to use the assumptions in ICA or BSS to derive Assumption 4, rather than “separability”, and the input of the algorithm is the data X, but not the mixing matrix.

6. How could we know the number of latent sources or the size of the mixing matrix?

I would raise my score if the authors could resolve all my concerns.


**Summary:**

This paper considers the causal discovery problem in linear structural causal models, where the observed variables are of a deterministic function of other observed variables or latent confounders, but not with an exogenous source.

---

> ### Author Response · Authors · 2021-12-03
> **Response to Reviewer fNiS**
>
> We thank the reviewer for the comments. We address the concerns below.
>
> 1. We first would like to clarify the definition of a latent confounder: Consider a setup in which a source variable $s$ is exogenously connected to more than one observed variable (say $x_1, x_2$). In this case, we can interpret $s$ as a distinct source for an *unobserved* variable, say $L$ (i.e., $s\rightarrow L \rightarrow (x_1, x_2)$), which is a latent confounder of $x_1$ and $x_2$. For example, in Figure 1(a), $s_3$ can be interpreted as $s_3 \rightarrow L$, where $L$ is a latent confounder of $x_1$ and $x_2$.
> The reviewer mentioned a modification of Figure 1(a), in which $s_1$ and $s_2$ are removed and a new source is added to $x_3$. We note that by our definition of latent confounders, in this modified model $x_1$ and $x_2$ will both be deterministic functions of the latent confounder $L$. Therefore, the fact that we have less sources than observed variables results in deterministically generated variables. We would like to emphasize that an observed variable is non-deterministic only if it is exogenously connected to distinct sources—sources that are not exogenously connected to any other observed variables. If deterministic relations do not exist in the system, then each observed variable must be associated with a distinct source, and therefore, the number of the sources must be more than or equal to the number of the observed variables.
>
> 2. In the special case of two-variable the reviewer is correct: If Assumption 2 is satisfied, then $x_j$ cannot be a deterministic function of $x_i$ or latent confounders (shared sources). However, when there are more than two observed variables, Assumption 2 does not always contradict non-determinism. One such example is shown in Figure 1(a). In this model, $x_3$ deterministically depends on its parents $x_1$ and $x_2$, but $x_3$ includes one more source ($s_2$) than $x_1$, and one more source ($s_1$) than $x_2$. Therefore Assumption 2 is satisfied and yet we have a deterministic relation.
>
>     Please note that, we believe the two-variable example with deterministic relation is a degenerate case, and there is no way to recover the causal direction between $x_i $ and $x_j$ only based on observations. Suppose there is a latent confounder $s_k$, and the generating model is: $x_i=b_1s_i+b_2 s_k$, $x_j=ax_i+b_3 s_k$. Then we can always construct an alternative model $x_j=ab_1 s_i+ (a b_2 + b_3) s_k$, $x_i=1/a~ x_j - b_3/a~ s_k$ with the same observational distributions of $x_i$ and $x_j$, where $x_j$ is a parent of $x_i$. In this case the true model cannot be distinguished from the alternative model by any method.
>
> 3. We thank the reviewer for the references. We will add these references in the revised paper.
>
> 4. We would like to clarify that the dimension of the mixing matrix $\mathbf{W}$ is $p\times m$, where $p$ is the number of observed variables, and $m$ is the number of sources. That is, we observe $p$ variables in $\mathcal{X}$, each of which is a linear combination of the $m$ independent sources in $\mathcal{S}$. Therefore, whether to use undercomplete ICA, standard ICA or overcomplete ICA for blind source separation depends on the number of the sources and the number of the observed variables, which further depends on the model structure. We note that even if deterministic relations exist in the system, there may still be more or equal sources than observed variables (such as Figure 1(a)).
> Further, we would like to clarify that undercomplete ICA is in fact included in standard ICA in most literature, as both can be solved using the same algorithm (e.g., FastICA).
>
> 5. The reason we assume separability instead of more specific assumptions on the model (such as non-Gaussianity of the sources which is assumed in LiNGAM) is that there are numerous models and corresponding BSS methods where the true mixing matrix can be correctly recovered. We do not want to restrict our recovery approach to a specific BSS model. For the input of the algorithm, we can add a black box for BSS and then change the input to observational data $X$.
>
> 6. It depends on the assumptions of the selected BSS method. In standard ICA there are always equal numbers of observed variables and latent sources. In undercomplete ICA and SBSSR model by Behr et al (2018), the number of latent sources can be recovered by additional statistical testings along with the recovery of the mixing matrix in the algorithm. In overcomplete ICA, the number of latent sources is always assumed to be known in advance.

---

> > ### Comment · Reviewer_fNiS · 2021-12-10
> > **Response to CLeaR 2022 Conference Paper105 Authors**
> >
> > Thanks for the authors’ responses. I have increased my score to 7.
> >
> > Following the authors’ responses, I have the following suggestions,
> >
> > i) it might be better to give a definition/explanation for deterministic relations, since some scholars may have thought that such deterministic relations are those $x=f(Pa(x))$ without noise (Glymour et al., 2019).
> >
> > ii) as for Assumption 2, it might be better to explain clearly that the model and/or theory does not hold for the two-variable cases.
> >
> > (Glymour et al., 2019) Glymour C, Zhang K, Spirtes P. Review of causal discovery methods based on graphical models[J]. Frontiers in genetics, 2019, 10: 524.

---

> > > ### Author Response · Authors · 2021-12-14
> > > **Thanks for the suggestions**
> > >
> > > We thank the reviewer for the suggestions. We will add both explanations in the revised paper.

---

### Official Review · Reviewer_P6YW · 2021-11-23

**Confidence:** 4
**Overall Score:** 8

**Main Review:**

I personally like the paper. It is clearly written and contains a clear result. The result and the algorithm, although marginally significant in my opinion, are novel and worth publishing. The technicalities seem correct to me although I did not check the Appendix carefully. A problem with the paper in current from is that a large number of examples, explanations, and numerical experiments that fit in the main body of the paper and help to understand the content are put in the Appendix. I think the authors could have chosen what goes into the main text and what goes to the Appendix better. I have just a few comments below:

At first, the paper sounds as though they have solutions for all possible deterministic relationships, but as it turns out later there are rather strong combinatorial conditions that need to be satisfied. I think the authors should be more clear about this from the beginning.

The argument about the faithfulness assumption being satisfied almost surely, although formally correct, is misleading. It is true that any additional conditional independence is of order zero, but any SCM assumes many of these order-zero constraints, so it is not totally unreasonable to wonder whether more of these independencies not lie around. In addition, even after looking at the related Appendix section, I could not see that it is shown formally that this condition is equivalent to the faithfulness assumption.

There should be more discussion on the main two assumption required in the main result. What are the causal interpretation and implications of these assumptions, and also how restrictive do they become?

It seems to me that the conditions could be used beyond the linear setting, as, in principle they do not use the assumption of linearity. It would be worthwhile to have a discussion on whether they generalize the known results on identifiability of non-linear SCMs.

**Summary:**

The paper generalizes the linear SCM setting to the case where simultaneously latent variables and deterministic relationships between endogenous variables are existent. The main result of the paper provides necessary and sufficient conditions for unique identifiability of causal effects among the variables. For the sufficiency direction, an algorithm to learn the causal structure is provided. It is shown that this algorithm recovers the true causal structure better than some of the alternative algorithms when dealing with some different subclasses of this general SCMs.

---

> ### Author Response · Authors · 2021-12-03
> **Response to Reviewer P6YW**
>
> We thank the reviewer for the comments and suggestions. We address the concerns below.
>
> **Regarding “strong combinatorial conditions need to be satisfied for identifiability”**:
>
> We would like to emphasize that the identifiability conditions in Section 3 are necessary and sufficient. Therefore, there cannot be any solution for the model studied in our paper with weaker identifiability conditions. We will emphasize these conditions at the beginning of the presentation in the revised paper. We thank the reviewer for this suggestion.
>
> **Faithfulness assumption in P-SCM**:
>
> P-SCM faithfulness assumption part (a) can be interpreted as: Any variable is marginally dependent on all its descendants. This is equivalent to the faithfulness assumption in Salehkaleybar et al. (2020) on linear SCMs with latent variables (DS-P-SCMs), and is in fact less restrictive than the common faithfulness assumption in studying causal models, which requires that any conditional independency in the distribution implies a corresponding d-separation in the causal diagram.
>
> Part (b) of the assumption is there to simply prevent the case when the coefficients of the exogenous connections from a subset of two or more sources to an observed variable is proportional to the coefficients of their exogenous connections to another observed variable. For example, suppose $x_1 = b_{11} s_1 + b_{21}s_2$ and $x_2 = b_{12} s_1 + b_{22}s_2$. Assumption (b) prevents $(b_{11}, b_{21})$ from being proportional to $(b_{12}, b_{22})$. This is not a restrictive assumption as if we were to introduce a continuous probability distribution over the non-zero entries of $\mathbf{B}$, part (b) merely rules out some zero probability events.
>
> We will add more explanation about the implications of the assumptions in the revised paper.
>
> **Regarding casual interpretation and restrictiveness of the conditions in main result**:
>
> The causal interpretation of unique component condition is that certain edges from source variables to observed variables are not allowed in the ground-truth structure. The interpretation of marriage condition is that the ancestor set of every observed variable has a sufficient number of exogenous connections (from source variables) so that it allows for unique identifiability of the causal effects to this observed variable. We will add these interpretations in the revised paper.
>
> As for the restrictiveness of the conditions, we provided experimental results in Section 5 in which we tested the satisfiability of the conditions for randomly generated P-SCMs and DS-P-SCMs. We evaluated what portion of them satisfy the conditions under different generating model assumptions, different number of causal/exogenous connections, observed variables, and source to observed node ratio. Please refer to Appendix G.1 for more details.

---

### Official Review · Reviewer_mdbo · 2021-11-25

**Confidence:** 3
**Overall Score:** 8

**Main Review:**

Originality: The identifiability results under determinism are new and relevant. The novelty compared to existing works is well justified.

Significance: The paper addresses an important problem, which is relevant for the CLeaR community. Determinism is for example implicit in constitutive relationships. Complex systems typically mix up constitutive and causal relationships. The paper for sure provides new insights. The insight of using blind source separation to find out deterministic relationships is interesting and full of potentiality. The paper is likely to have broad impacts also outside the CLeaR community, in all complex settings where causal and constitutive relationships are mixed up.

Technical quality: The proposed approach is technically sound, with claims substantiated by sound theoretical results. However, some the implications of the so-called "unique components condition" should be better spelled out. It seems to me that this condition implies that one observes all the variables that enter as argument in a deterministic function of the structural causal model. Suppose that, in the SCM, X=f(Y,Z). It seems to me that the unique component condition (and the proposed identification method to function) needs that one observed Y and Z. Latent confounding is allowed only when is nondeterministic. I think this should be spelled out. Moreover, at p. 6 (bottom) it is written: "we can use this unique component to compute the total causal effect from $x_i$ to $x_k$ (by dividing the coefficient of this unique component in $x_k$ by its coefficient in $x_i$)". What is claimed between brackets should be rigorously proved.

The paper is clearly written and well organised. However, in the introduction section some illustrative examples of the different SCMs would help.

**Summary:**

The main contribution of the paper to the literature is to provide identifiability results for causal discovery under the assumption of linearity and allowing for both latent confounding and deterministic relationships.

---

> ### Author Response · Authors · 2021-12-03
> **Response to Reviewer mdbo**
>
> We thank the reviewer for the comments and we are glad that the reviewer finds our work insightful. We address the concerns below.
>
> **Regarding the comment about “observing all the variables that enter as argument in a deterministic function of the structural causal model”**:
>
> We would like to clarify that we only observe the “observed variables” in $\mathcal{X}$, not the source variables. All source variables in $\mathcal{S}$ are exogenous and are not observed. Recall that in our setup we define a latent confounder as a source variable $s$ that is exogenously connected to more than one observed variable (say $x_i$ and $x_j$). In this case, we can interpret $s$ as a distinct source for an *unobserved* variable, say $L$ (i.e., $s\rightarrow L$ with generating model $L=s$), which is a latent confounder of the corresponding observed variables (i.e., replacing the edges $s\rightarrow (x_i,x_j)$ by $L \rightarrow (x_i, x_j) $).
>
> For example, consider the structure in Figure 1(a), with an extra edge from $s_3$ to $x_3$ added to the graph. Therefore, $s_3$ (or equivalently the added child, $L$) is a confounder of all $x_1$, $x_2$, and $x_3$, and $x_3=f(x_1, x_2, L)$. First, the structure of $x_3$ in this model satisfies the unique component condition, which simply requires that there should not be any edges from $s_1$ to $x_3$ and from $s_2$ to $x_3$. Second, one argument of the function that generates $x_3$ is not observed, i.e., $L$. Third, $x_3$ is a deterministic function of $x_1$, $x_2$ and latent confounder $L$. Therefore, this is an example in which (i) we have a latent confounder (shared source) in the system, (ii) an observed variable is deterministically generated from other observed variables and latent confounders, and (iii) the generating model does satisfy our unique component condition. Also, this structure can be uniquely recovered by our proposed algorithm.
>
> **Regarding the statement that we compute the total causal effect from $x_i$ to $x_k$ by dividing the coefficients**:
>
> The rigorous proof is included in the sufficiency proof in Appendix F.2, and we further explained this claim in Remark 9 (in the proof), Page 30. We will clarify it further in the text and will refer to the proof in the revised paper.
>
> **Illustrative examples of the different SCMs**:
>
> We thank the reviewer for this suggestion regarding improving the presentation. We will add more illustrative examples in the introduction.

---

### Decision · Program_Chairs · 2022-01-13

**Decision:**

Accept (Poster)

**Comment:**

The paper studies causal structure learning of linear SCMs under confounding, while allowing for deterministic relationships in the structural equations. The authors derive necessary and sufficient conditions for identifiability and provide an algorithm to recover the structure. The reviewers agree that the results are an interesting contribution to the literature. That being said, there are minor issues in the exposition that should be addressed in the final revision (see reviewer fNIS’s suggestions). In addition, it would be great if the authors could provide more intuition on the two major assumptions in the main paper (e.g., by providing examples).